# Approximate Bayesian Inference with Stein Functional Variational Gradient Descent

**Tobias Pielok, Bernd Bischl, David Rügamer**
Department of Statistics, LMU Munich, Munich, Germany
Munich Center for Machine Learning, Munich, Germany
`{tobias.pielok, bernd.bischl, david.ruegamer}@stat.uni-muenchen.de`

## Abstract

We propose a general-purpose variational algorithm that forms a natural analogue of Stein variational gradient descent (SVGD) in function space. While SVGD successively updates a set of particles to match a target density, the method introduced here of Stein functional variational gradient descent (SFVGD) updates a set of particle functions to match a target stochastic process (SP). The update step is found by minimizing the functional derivative of the Kullback-Leibler divergence between SPs. SFVGD can either be used to train Bayesian neural networks (BNNs) or for ensemble gradient boosting. We show the efficacy of training BNNs with SFVGD on various real-world datasets.

## 1 Introduction

Bayesian inference can be treated as a powerful framework for data modeling and reasoning under uncertainty. However, this assumes that we can encode our prior knowledge in a meaningful manner. Typically, this is done by specifying the prior distribution of the model parameters. However, in machine learning (ML), models potentially consist of millions of parameters with potentially highly complex interactions (e.g., very large neural networks (NNs)). Furthermore, the parameter structure of the models itself is allowed to change during training, e.g., the number of parameter grows when using gradient boosting (GB). This makes defining meaningful prior assumptions for parameter spaces difficult or nearly (practically) infeasible. As we usually do not care about single parameters but the complete resulting function, it seems intuitive to directly express our prior knowledge in hypothesis function space by, e.g., specifying the characteristic length scale, periodicity, or smoothness in general. Fortunately, Bayesian inference can also be formulated in function space. In this case, the prior and posterior distributions are stochastic processes (SPs). The most prominent representative is the Gaussian process (GP), for which the posterior GP can be analytically computed. However, training GPs scale cubically in the number of observations, and the implicit Gaussian likelihood assumption is often violated in reality. In this paper, we introduce Stein functional variational gradient descent (SFVGD). This method provides a general gradient descent method in function space that enables practitioners to train ML models to approximate the posterior SP, assuming certain regularity conditions of the prior SP and the likelihood function hold.

### 1.1 Related work

**Kernelized Stein Methods** These methods combine Stein's identity with a reproducing kernel Hilbert space (RKHS) assumption. Based on a finite particle set, they can either be used to find the optimal transport direction to match a target density or to estimate the score gradient of the empirical distribution of the particles. The former is called Stein variational gradient descent (SVGD) Liu & Wang (2016), and approaches of the latter category are called (non-parametric) score estimators (Zhou et al., 2020). Our method internally uses SVGD and forms a natural analogue in function space. Several extensions to SVGD exist, e.g., approaches incorporating second-order information such as Leviyev et al. (2022) and the more general matrix-kernel valued approach by Wang et al. (2019a). While these extensions usually outperform SVGD, their computational costs are also higher.

**Bayesian Neural Networks (BNNs)** Typically, BNNs are NNs with weight priors that are trained via variational inference. The prominent representatives are BNNs using *Bayes by Backprop* (Blundell

et al., 2015) and scalable probabilistic backpropagation (Hernandez-Lobato & Adams, 2015). Recently, Immer et al. (2020) proposed transforming BNNs into generalized linear models with inference based on a Gaussian process equivalent to the model. While Markov Chain Monte Carlo (MCMC) methods often are prohibitively expensive to be used for BNNs, some variants, e.g., Chen et al. (2014) account for noisy gradient evaluations and can be used in this setting. However, MCMC-based methods are still usually employed for relatively low-dimensional problems.

**Functional BNNs (FBNNs)** Sun et al. (2019) proposed to use functional priors to train BNNs. Training BNNs with our descent is closely related to their method, but while they use a score-based approach for the estimation of the derivative of the Kullback-Leibler divergence $D_{\mathrm{KL}}$ between the prior SP and the variational SP, we estimate this derivative directly via SVGD. Wang et al. (2019b) also use SVGD for FBNNs but apply SVGD directly to the $D_{\mathrm{KL}}$ between the posterior SP and the variational SP. Furthermore, their work does not show that this in fact maximizes a lower bound for the log marginal likelihood. Recently, Ma & Hernández-Lobato (2021) and Rudner et al. (2021) proposed different FBNN approaches that also build upon the results of Sun et al. (2019), but while their methods are specific to training NN function generators, our method can be used to update a set of particle functions in general.

**Repulsive Deep Ensembles** Repulsive Deep Ensembles are deep ensembles that incorporate repulsive terms in their gradient update, forcing their members' weights apart. A variety of repulsive terms are presented in (D'Angelo & Fortuin, 2021) and (D'Angelo et al., 2021), outperforming the approach by Wang et al. (2019b). However, these approaches mainly focus on weight priors, and empirical findings also only relate to the weight space. In contrast to our work, functional priors can only be applied if a posterior SP with analytical marginal density exists.

**GB with Uncertainty** The closest neighbor of our approach applied to GB is the ensemble GB scheme proposed by Malinin et al. (2021), which is based on Bayesian ensembles. In contrast to our functional approach, their method is based on approximating the posterior of the model parameters. Another GB-based method is NGBoost proposed by Duan et al. (2019), which directly learns the predictive uncertainty; however, prior knowledge can not be taken into account.

## 1.2 OUR CONTRIBUTION

We propose a novel natural extension of SVGD in function space (Section 3), which enables the practitioner to match a target SP. This approach can be implemented in a BNN or as GB routine (Section 3.3). Using real-world benchmarks, we show that the resulting generator training algorithm is competitive while having less computational costs than the approach of Sun et al. (2019). In contrast to other existing uncertainty-aware GB algorithms, a GB ensemble, when trained via SFVGD, can naturally incorporate prior functional information. These versatile applications of our framework are made possible by providing a unifying view of NNs and GB from a functional analysis perspective.

## 2 BACKGROUND

### 2.1 SUPERVISED ML FROM A FUNCTIONAL ANALYSIS PERSPECTIVE

Given a labeled dataset $\mathcal{D} \in (\mathcal{X} \times \mathcal{Y})^n \sim \mathbb{P}_{\mathbf{x}y}^n$ of $n \in \mathbb{N}$ independent and identically distributed (i.i.d.) observations from an unknown data generating process $\mathbb{P}_{\mathbf{x}y}$, a supervised ML algorithm tries to construct a risk optimal model $f$ under a pre-specified loss $L$. In this case, the function $f$ defines a mapping from the feature space $\mathcal{X}$ to the target space $\mathcal{Y}$. The learning algorithm $\mathcal{I}$ to construct $f$ is a function mapping from the set of all datasets $\bigcup_{n \in \mathbb{N}} (\mathcal{X} \times \mathcal{Y})^n$ to a hypothesis space $\mathcal{H}$, which is a subset of the set of all functions mapping from $\mathcal{X}$ to the model output space[1] $\widetilde{\mathcal{Y}} \subset \mathbb{R}^g$ with $g \in \mathbb{N}$. In order to specify the goodness-of-fit of a function $f$, one can define a loss function $L : \mathcal{Y} \times \widetilde{\mathcal{Y}} \to \mathbb{R}, (y, f(\mathbf{x})) \mapsto L(y, f(\mathbf{x}))$, which measures how well the output of a fixed model $f \in \mathcal{H}$ fits an observation $(\mathbf{x}, y) \sim \mathbb{P}_{\mathbf{x}y}$. In the following, we present supervised ML from a functional analysis perspective. Here, we fix the observation and associate the loss $L$ with the loss functional $L_{(\mathbf{x},y)}[f] : \mathcal{H} \to \mathbb{R}, f \mapsto L(y, f(\mathbf{x}))$. Based on this loss functional, we can define the risk functional of a model $f$,

$$\mathcal{R}[f] = \mathbb{E}_{(\mathbf{x},y) \sim \mathbb{P}_{\mathbf{x}y}} L_{(\mathbf{x},y)}[f], \tag{1}$$

---

[1]If $\mathcal{Y}$ is numeric, $\widetilde{\mathcal{Y}} = \mathcal{Y}$. Otherwise, $\widetilde{\mathcal{Y}}$ is a numerical encoding of $\mathcal{Y}$.

which measures the expected loss of $f$, and is used to theoretically identify optimal models. In the following, we will assume that the expectation in Eq. (1) exists and is finite. If we knew the usually unknown data generating process and hence the risk functional, we could update any model $f \in \mathcal{H}$ in the direction of the steepest descent in $\mathcal{H}$ w.r.t. $\mathcal{R}$ by following the negative functional gradient of $\mathcal{R}$. The negative functional gradient of $\mathcal{R}$, $-\nabla_f \mathcal{R}[f]$, is itself a mapping from $\mathcal{X}$ to $\widetilde{\mathcal{Y}}$. For every input location $\mathbf{x}$, this gradient returns the direction in model output space $\widetilde{\mathcal{Y}}$, which points to the locally steepest descent w.r.t. $\mathcal{R}$. In the following, unless otherwise stated, the functional derivative is taken in the $L^2$ space.

**Proposition 2.1** *Assuming sufficient regularity and that $L(y, f(\mathbf{x}))$ is partially continuously differentiable w.r.t. $f(\mathbf{x})$, we observe for numeric inputs and model output that*

$$- \nabla_f \mathcal{R}[f](\mathbf{x}) = -p_{\mathbf{x}}(\mathbf{x}) \cdot \mathbb{E}_{y \sim \mathbb{P}_{y|\mathbf{x}}} \frac{\partial L(y, f(\mathbf{x}))}{\partial f(\mathbf{x})}, \tag{2}$$

*where $p_{\mathbf{x}}$ is the marginal density of $\mathbf{x}$.*

The proof is given in A.1.1. In practice, we usually do not know $p_{\mathbf{x}}$, and since our dataset $\mathcal{D}$ is finite, we only have access to $n$ realizations of $\frac{\partial L(y, f(\mathbf{x}))}{\partial f(\mathbf{x})}$. If the feature space is at least partially continuous, its size $|\mathcal{X}| = \infty$, and we thus cannot estimate $-\nabla_f \mathcal{R}[f](\mathbf{x})$ without additional assumptions. However, we have access to the functional empirical risk $\mathcal{R}_{\text{emp},\mathcal{D}}[f] := \sum_{(\mathbf{x}_i, y_i) \in \mathcal{D}} L_{(\mathbf{x}_i, y_i)}[f]$, for which we assume that it converges in mean to $\mathcal{R}$ as $n \to \infty$. Its negative functional gradient can be expressed via the chain rule such that

$$- \nabla_f \mathcal{R}_{\text{emp},\mathcal{D}}[f](\mathbf{x}) = - \sum_{(\mathbf{x}_i, y_i) \in \mathcal{D}} \frac{\partial L(y_i, f(\mathbf{x}_i))}{\partial f(\mathbf{x}_i)} \cdot \nabla_f [f(\mathbf{x}_i)](\mathbf{x}), \tag{3}$$

where $\nabla_f [f(\mathbf{x}_i)]$ is the functional gradient of the evaluation functional of $f$ at $\mathbf{x}_i$, which evaluates to the Dirac delta function $\delta_{\mathbf{x}_i}$. However, since we take the functional gradient in $\mathcal{H}$, $\nabla_f [f(\mathbf{x}_i)]$ becomes the projection of $\delta_{\mathbf{x}_i}$ into $\mathcal{H}$. For example, if $\mathcal{H}$ is an RKHS with associated kernel $k$, then $\nabla_f [f(\mathbf{x}_i)](\mathbf{x}) = k(\mathbf{x}_i, \mathbf{x})$, i.e., our choice of $\mathcal{H}$ directly influences the "bumpiness" of $\nabla_f \mathcal{R}_{\text{emp},\mathcal{D}}[f]$. Furthermore, we can interpret $\nabla_f \mathcal{R}_{\text{emp},\mathcal{D}}[f]$ as a (jump-)continuous functional representation of the dataset $\partial \mathcal{D}_{L,f} := \{(\mathbf{x}_i, -\frac{\partial L(y_i, f(\mathbf{x}_i))}{\partial f(\mathbf{x}_i)}) \mid (\mathbf{x}_i, y_i) \in \mathcal{D}\} \subset (\mathcal{X} \times \widetilde{\mathcal{Y}})^n$, which also implicitly defines a learner. In the following, we show how two core supervised ML algorithms (gradient boosting and neural networks) naturally incorporate this functional gradient while training.

**Gradient Boosting (GB)** For GB (Friedman, 2001), the situation is usually reversed, and we choose a (base) learner $\mathcal{I}_b$ that implicitly defines $\mathcal{H}$ and with which we fit a model to the data set $\partial \mathcal{D}_{L,f}$. GB uses these approximations of the negative functional gradient of the empirical risk to successively update a model $f^{[0]}$ such that

$$f^{[t+1]} = f^{[t]} + \eta^{[t]} b^{[t]} \text{ with } b^{[t]} = \mathcal{I}_b(\partial \mathcal{D}_{L, f^{[t]}}), \tag{4}$$

where $\eta^{[t]} \in \mathbb{R}_{>0}$ is the learning rate and possibly depends on the iteration $t \in \mathbb{N}$. For further details see Appendix (A.2).

**Neural Networks (NNs)** If $f$ is an NN with parameters $\phi$, then the parameter gradients w.r.t. the empirical risk functional needed for backpropagation can be obtained via the chain rule such that

$$\nabla_\phi \mathcal{R}_{\text{emp},\mathcal{D}}[f] = \int_{\mathcal{X}} \nabla_f \mathcal{R}_{\text{emp},\mathcal{D}}[f](\mathbf{x}) \cdot \nabla_\phi f(\mathbf{x}) \mathrm{d}\mathbf{x} = \sum_{(\mathbf{x}_i, y_i) \in \mathcal{D}} \frac{\partial L(y_i, f(\mathbf{x}_i))}{\partial f(\mathbf{x}_i)} \cdot \nabla_\phi f(\mathbf{x}_i), \quad (5)$$

where the second equality holds, since here we do not restrict $\mathcal{H}$, i.e., $\nabla_f [f(\mathbf{x}_i)] = \delta_{\mathbf{x}_i}$.

However, these procedures only assure that we can find an optimal model $f \in \mathcal{H}$ w.r.t. $\mathcal{R}_{\text{emp}}$, which does not imply that $f$ is optimal w.r.t. $\mathcal{R}$. In practice, we tune the hyperparameters of the algorithms – i.e., use data withheld from learning for subsequent model selection – and apply early stopping to find a model $f$ approximately optimal w.r.t. $\mathcal{R}$.

## 2.2 STOCHASTIC PROCESSES

In this section, we will shortly introduce stochastic processes (SPs) that can be used to represent distributions over functions and thereby allow us to express the uncertainty of models independent of their specific parameter structure. We will regard $\mathcal{X}$ as an index set and let $(\mathcal{Y}, \mathcal{G})$ be a measurable space with $\sigma$-algebra $\mathcal{G}$ on the state space $\mathcal{Y}$. For $\mathbf{x} \in \mathcal{X}$ and a given probability space $(\Omega, \mathcal{F}, \mathbb{P})$ based on the sample space $\Omega$, $\mathcal{F}$ is a $\sigma$-algebra on $\Omega$ and probability measure $\mathbb{P}$. Let $Q(\mathbf{x})$ be a random variable projecting from $\Omega$ to $\mathcal{Y}$. An SP $Q$ is the family $\{Q(\mathbf{x}); \mathbf{x} \in \mathcal{X}\}$ of all random variables $Q(\mathbf{x})$ (Lamperti, 1977). With this, we can define a sample function $f_\omega : \mathcal{X} \to \mathcal{Y}, \mathbf{x} \mapsto Q(\mathbf{x})(\omega)$ for a fixed $\omega \in \Omega$. Often, it is easier to look at SPs from this sample function view: For every $A \in \mathcal{F}$, a set of functions $\{f_\omega; \omega \in A\}$ with an associated measure $\mathbb{P}(A)$ can be identified – i.e., SPs define a distribution over functions projecting from $\mathcal{X}$ to $\mathcal{Y}$. For a finite index set $\mathbf{X} := \mathbf{x}_{1:m} \in \mathcal{X}^m$, we denote the finite-dimensional marginal joint distribution over function values $\{Q(\mathbf{x}_1), \ldots, Q(\mathbf{x}_m)\}$ as $Q_{\mathbf{X}}$. In the following, we assume that for every $Q_{\mathbf{X}}$ exists a corresponding density function $p_{Q_{\mathbf{X}}} : \mathcal{Y}^m \to \mathbb{R}_{\geq 0}, \mathbf{f}^{\mathbf{X}} \mapsto p_{Q_{\mathbf{X}}}(\mathbf{f}^{\mathbf{X}})$, where $\mathbf{f}^{\mathbf{X}} := (f(\mathbf{x}_1), \ldots, f(\mathbf{x}_m))$ are the function values at $\mathbf{x}_{1:m}$ based on a sample function $f$ where we suppressed the $\omega$ to ease the following notation. We will denote the associated functional to this density function with $p_{Q_{\mathbf{X}}}[f]$.

The $D_{\mathrm{KL}}$ is a measure of distance between two distributions over the same probability space. Since SPs are distributions over functions, the $D_{\mathrm{KL}}$ can also be used for distances between two SPs. Unfortunately, computing this quantity is non-trivial (Matthews et al., 2015). However, for two consistent and ergodic SPs $Q$ and $P$, i.e., $Q$ and $P$ can be characterized by marginals over all finite index sets (e.g., GPs), Sun et al. (2019) showed that the $D_{\mathrm{KL}}$ between these SPs can be solely expressed in terms of their marginals, i.e.,

$$D_{\mathrm{KL}}(Q\|P) = \sup_{m \in \mathbb{N}, \mathbf{X} \in \mathcal{X}^m} D_{\mathrm{KL}}(Q_{\mathbf{X}}\|P_{\mathbf{X}}). \tag{6}$$

This expression enables us to find a differentiable distance measure between two stochastic processes.

## 2.3 STEIN VARIATIONAL GRADIENT DESCENT

SVGD (Liu & Wang, 2016) is a variational Bayesian method. Variational methods can be used to approximate the generally intractable posterior density of a continuous random variable $\boldsymbol{\theta}$

$$p_{\theta|\mathcal{D}}(\boldsymbol{\theta}) = \frac{p_{\mathcal{D}|\theta}(\mathcal{D}|\boldsymbol{\theta})p_\theta(\boldsymbol{\theta})}{\int p_{\mathcal{D}|\theta}(\mathcal{D}|\boldsymbol{\theta})p_\theta(\boldsymbol{\theta})\mathrm{d}\boldsymbol{\theta}}, \tag{7}$$

where $p_{\mathcal{D}|\theta}$ and $p_\theta$ are the likelihood and the prior density function, respectively. SVGD tries to match the posterior $p_{\theta|\mathcal{D}}$ with a density $q$ represented via a fixed number $r \in \mathbb{N}$ of pseudo-samples – so-called particles – and iteratively updates them by minimizing $D_{\mathrm{KL}}(q\|p_{\theta|\mathcal{D}}) = \int q(\boldsymbol{\theta})\frac{\log(q(\boldsymbol{\theta}))}{p_{\theta|\mathcal{D}}(\boldsymbol{\theta})}d\boldsymbol{\theta}$. In an RKHS with associated kernel $k$, the optimal update direction is found by considering the negative functional derivative

$$-\nabla_f D_{\mathrm{KL}}(q_{[T]}\|p_{\theta|\mathcal{D}})\big|_{f=0} = \mathbb{E}_{\theta \sim q}\left[\nabla_\theta \log p_{\theta|\mathcal{D}}(\boldsymbol{\theta})k(\boldsymbol{\theta}, \cdot) + \nabla_\theta k(\boldsymbol{\theta}, \cdot)\right], \tag{8}$$

where $T(\boldsymbol{\theta}) = \boldsymbol{\theta} + f(\boldsymbol{\theta})$, and $q_{[T]}$ is the density of $\boldsymbol{\theta}' = T(\boldsymbol{\theta})$ when $\boldsymbol{\theta} \sim q$. We can estimate this functional gradient based on the particles in an unbiased manner, as we are able to evaluate the score function of $p_{\theta|\mathcal{D}}$ (i.e., $\nabla_\theta \log p_{\theta|\mathcal{D}}$), although $\log p_{\theta|\mathcal{D}}$ might be intractable.

# 3 STEIN FUNCTIONAL VARIATIONAL GRADIENT DESCENT

In this section, we develop a functional version of SVGD which we will call *Stein functional variational gradient descent* (SFVGD). While SVGD can be used to approximate the posterior distribution of a continuous random variable, SFVGD can be applied when we are interested in the posterior SP $P_{f|\mathcal{D}}$ defined by its Radon-Nikodym derivative (Schervish, 1995) w.r.t. the prior SP $P_f$,

$$\frac{dP_{f|\mathcal{D}}}{dP_f}[f] = \frac{p_{\mathcal{D}|f}[\mathcal{D}|f]}{\int p_{\mathcal{D}|f}[\mathcal{D}|f]dP_f[f]}, \tag{9}$$

where $p_{\mathcal{D}|f}$ is the likelihood functional, which measures how likely it is to observe $\mathcal{D}$, given a sample function $f$. In the following, we assume that the posterior $P_{f|\mathcal{D}}$ exists and also that it is an ergodic and consistent SP. Analogously to SVGD, we try to approximate $P_{f|\mathcal{D}}$ with a distribution $Q$ represented by pseudo-samples. However, for SFVGD, these particles are now functions.

### 3.1 OBJECTIVE FUNCTION

Since analytical solutions for the differential Eq. (9) only exist in special cases (e.g., if the prior $P_f$ is a GP and the likelihood is also Gaussian), we use the $D_{\mathrm{KL}}$ between two SPs to formulate an optimization objective. More specifically, the goal of our framework is to construct an approximating measure $Q^*$ for which it holds that

$$Q^* \in \arg\min_{Q \in \mathcal{Q}} D_{\mathrm{KL}}(Q\|P_{f|\mathcal{D}}), \tag{10}$$

where $\mathcal{Q}$ is the set of representable variational posterior processes. Here, we represent $Q$ via $r \in \mathbb{N}$ sample functions $f_1, \ldots, f_r$ from $Q$, which act as pseudo-samples and which we also call particle functions. It can be shown (Matthews et al., 2015) that minimizing Eq. (10) is equivalent to maximizing the functional evidence lower bound (ELBO) $\mathcal{L}_{\mathcal{D}}$, i.e.,

$$Q^* \in \arg\max_{Q \in \mathcal{Q}} \underbrace{\mathbb{E}_{f \sim Q}\left[\ell[\mathcal{D}|f]\right] - D_{\mathrm{KL}}(Q\|P_f)}_{=:\mathcal{L}_{\mathcal{D}}(Q)}, \tag{11}$$

where $\ell[\mathcal{D}|f] := \log p_{\mathcal{D}|f}[\mathcal{D}|f]$. The advantage of formulation (11) over (10) is that Eq. (11) only depends on known quantities. In the following, we apply Eq. 6, i.e., the results of Sun et al. (2019) regarding the $D_{\mathrm{KL}}$ of ergodic and consistent SPs, yielding

$$Q^* \in \arg\max_{Q \in \mathcal{Q}} \inf_{m \in \mathbb{N}, \mathbf{X} \in \mathcal{X}^m} \underbrace{\mathbb{E}_{f \sim Q}[\ell[\mathcal{D}|f]] - D_{\mathrm{KL}}(Q_{\mathbf{X}}\|P_{f\mathbf{X}})}_{=:\mathcal{L}_{\mathcal{D},\mathbf{x}}(Q)}. \tag{12}$$

In contrast to Sun et al. (2019), however, we do not unfold the $D_{\mathrm{KL}}$ term, since we are able to directly take its functional gradient via SVGD. The resulting maximin game formulation of Eq. (12) proves to be challenging to solve, especially since we need to minimize over discrete sets $\mathbf{X}$ and the infimum also does not ensure a finite $m$. Hence, we follow Sun et al. (2019) by replacing the inner minimization with a sampling-based approach, i.e.,

$$Q^* \in \arg\max_{Q \in \mathcal{Q}} \mathbb{E}_{\mathcal{D}_s} \mathbb{E}_{\mathbf{X}_M \sim C_{\mathcal{X}}} \left[\mathcal{L}_{\mathcal{D}_s,[\mathbf{X}_{\mathcal{D}_s},\mathbf{X}_M]}(Q)\right], \tag{13}$$

where $\mathcal{D}_s$ is a random subsample of size $|\mathcal{D}_s| = s$ drawn from $\mathcal{D}$. $\mathbf{X}_{\mathcal{D}_s}$ are the associated feature vectors of $\mathcal{D}_s$, and $\mathbf{X}_M = [\mathbf{x}_1, \ldots, \mathbf{x}_M]^\top \in \mathcal{X}^M$ are $M$ stacked random feature vectors drawn from a sampling distribution $C_{\mathcal{X}}$ with support $\mathcal{X}$. If $\mathcal{X}$ is bounded, Sun et al. (2019) proposes a uniform distribution for $C_{\mathcal{X}}$. It has been shown in Sun et al. (2019) for $\mathcal{D}_s = \mathcal{D}$ and $M > 1$ that $\mathcal{L}_{\mathcal{D},[\mathbf{X}_{\mathcal{D}},\mathbf{X}_M]}$ is a lower bound for the log marginal likelihood $\log p(\mathcal{D})$, i.e., the maximization in Eq. 13 implies the minimization in Eq. 10. Although, as noted by Burt et al. (2020), if $\mathcal{Q}$ is a parametric family, the objective is ill-defined, we did not encounter any problems in practice. Also, we could straightforwardly use the grid functional $D_{\mathrm{KL}}$ proposed by Ma & Hernández-Lobato (2021), which fixes some of these theoretical shortcomings. However, note that SFVGD itself does not assume $\mathcal{Q}$ to be parametric.

### 3.2 FUNCTIONAL DERIVATIVE OF THE OBJECTIVE

When using conventional gradient descent methods, we want to apply a map to update the parameters of our model such that our loss is reduced. In SFVGD, we proceed in a similar manner but update functions towards a loss-minimizing direction. A map that takes a function as an argument and returns another function is called an operator. Hence, we want to express how our objective value Eq. 13 changes when an operator $F : \mathcal{H} \to \mathcal{H}, f \mapsto f_F$ is applied to every $f \sim Q$. This means that the objective value changes with $F$ such that

$$\mathcal{L}_{\mathcal{D}_s,\mathbf{X}}(Q_{[T]}) = \mathbb{E}_{\widetilde{f} \sim Q_{[T]}} \ell[\mathcal{D}_s|f] - D_{\mathrm{KL}}(Q_{[T]\mathbf{X}}\|P_{f\mathbf{X}}), \tag{14}$$

where $T(f) = f + F(f)$ and $Q_{[T]}$ is the distribution of $\widetilde{f} = T(f)$ when $f \sim Q$. Naturally, we are interested in the functional derivative of Eq. 14 w.r.t. to $F$, since this gives us the direction of the steepest ascent in operator space regarding the functional ELBO. However, in order to make our computations tractable, we must limit the space of feasible operators:

**Definition 3.1** *Let $F : \mathcal{H} \to \mathcal{H}, f \mapsto f_F$ be a continuous operator with the property that for all $m \in \mathbb{N}$ and each $\mathbf{X} \in \mathcal{X}^m$ exists a function $F_{\mathbf{X}} : \mathcal{Y}^m \to \mathcal{Y}^m$ such that $\mathbf{f}_F^{\mathbf{X}} = F_{\mathbf{X}}(\mathbf{f}^{\mathbf{X}})$ for any $f \in \mathcal{H}$. We call such an operator "evaluation-only dependent".*

Thus, $F$ does not depend on derivatives of $f$ (which is not a restriction, since we only assumed $f$ to be continuous); we can also treat $F$ as a construction rule of $F_{\mathbf{X}}$ for arbitrary $m$ and $\mathbf{X}$. Now, we can state the functional gradient of the objective functional w.r.t. an evaluation-only dependent operator $F$, for $\mathbf{X} = [\mathbf{X}_{\mathcal{D}_s}, \mathbf{X}_M]$ and $\widetilde{\mathbf{X}} = \mathbf{X}_{\mathcal{D}_s}$

$$
\begin{aligned}
\nabla_F \mathcal{L}_{\mathcal{D}_s, \mathbf{X}}(Q_{[T]}) &= \nabla_F \mathbb{E}_{\widetilde{f} \sim Q_{[T]}} \ell[\mathcal{D}_s | \widetilde{f}] - \nabla_F D_{\mathrm{KL}}(Q_{[T]\mathbf{X}} \| P_{f\mathbf{X}}) \\
&= \nabla_F \mathbb{E}_{\widetilde{\mathbf{y}} \sim Q_{[T]\widetilde{\mathbf{X}}}} \ell(\mathcal{D}_s, \widetilde{\mathbf{y}}) - \nabla_F D_{\mathrm{KL}}(Q_{[T]\mathbf{X}} \| P_{f\mathbf{X}}),
\end{aligned}
\tag{15}
$$

where we assumed that there exists a log-likelihood function $\ell : \bigcup_{s \in \mathbb{N}} (\mathcal{X} \times \mathcal{Y})^s \times \mathcal{Y}^s \to \mathbb{R}$ such that $\ell[\mathcal{D}_s | f] = \ell(\mathcal{D}_s, \mathbf{f}^{\widetilde{\mathbf{X}}})$ for every $\mathcal{D}_s$. If we set $F = 0$, then $T$ becomes the identity operator, i.e. $Q_{[T]} = Q$. Since we want to iteratively update our particle functions, we must only consider small perturbations around $F = 0$.

**Proposition 3.1** *For an evaluation-only dependent operator $F$, the functional derivative of the functional ELBO at $F = 0$ evaluated for a function $f$*

$$
\begin{aligned}
\nabla_F \mathcal{L}_{\mathcal{D}_s, \mathbf{X}}(Q_{[T]})\big|_{F=0}(f) = {}& \mathbb{E}_{\widetilde{\mathbf{y}} \sim Q_{\widetilde{\mathbf{X}}}} \left[ \nabla_{\widetilde{\mathbf{y}}} \ell(\mathcal{D}_s, \widetilde{\mathbf{y}}) \cdot \delta_{\widetilde{\mathbf{y}}}(\mathbf{f}^{\widetilde{\mathbf{X}}}) \right] \cdot \left[ \delta_{\widetilde{\mathbf{X}}_1}(\cdot), \dots, \delta_{\widetilde{\mathbf{X}}_s}(\cdot) \right]^{\top} \\
&+ \mathbb{E}_{\mathbf{y} \sim Q_{\mathbf{X}}} \left[ \nabla_{\mathbf{y}} \log p_{P_{f\mathbf{X}}}(\mathbf{y}) k_{\mathbf{Y}}(\mathbf{y}, \mathbf{f}^{\mathbf{X}}) + \nabla_{\mathbf{y}} k_{\mathbf{Y}}(\mathbf{y}, \mathbf{f}^{\mathbf{X}}) \right] \\
&\cdot \left[ \delta_{\mathbf{X}_1}(\cdot), \dots, \delta_{\mathbf{X}_{s+M}}(\cdot) \right]^{\top},
\end{aligned}
\tag{16}
$$

*where we assume that $\mathcal{H}_{\mathbf{Y}} \subset \{f : \mathcal{Y}^{s+M} \to \mathcal{Y}^{s+M}\}$ is an RKHS with associated kernels $k_{\mathbf{Y}}$.*

The proof is given in A.1.2, where we also show the following corollary.

**Corollary 3.1.1** *For an evaluation-only dependent operator $F$, the functional derivative of the functional ELBO at $F = 0$ evaluated for a function $f$*

$$
\begin{aligned}
\nabla_F \mathcal{L}_{\mathcal{D}_s, \mathbf{X}}(Q_{[T]})\big|_{F=0}(f) = {}& \mathbb{E}_{\widetilde{\mathbf{y}} \sim Q_{\widetilde{\mathbf{X}}}} \left[ \nabla_{\widetilde{\mathbf{y}}} \ell(\mathcal{D}_s, \widetilde{\mathbf{y}}) \cdot k_{\widetilde{\mathbf{Y}}}(\widetilde{\mathbf{y}}, \mathbf{f}^{\widetilde{\mathbf{X}}}) \right] \cdot \left[ \delta_{\widetilde{\mathbf{X}}_1}(\cdot), \dots, \delta_{\widetilde{\mathbf{X}}_s}(\cdot) \right]^{\top} \\
&+ \mathbb{E}_{\mathbf{y} \sim Q_{\mathbf{X}}} \left[ \nabla_{\mathbf{y}} \log p_{P_{f\mathbf{X}}}(\mathbf{y}) k_{\mathbf{Y}}(\mathbf{y}, \mathbf{f}^{\mathbf{X}}) + \nabla_{\mathbf{y}} k_{\mathbf{Y}}(\mathbf{y}, \mathbf{f}^{\mathbf{X}}) \right] \\
&\cdot \left[ \delta_{\mathbf{X}_1}(\cdot), \dots, \delta_{\mathbf{X}_{s+M}}(\cdot) \right]^{\top},
\end{aligned}
\tag{17}
$$

*where we assume that $\mathcal{H}_{\mathbf{Y}} \subset \{f : \mathcal{Y}^{s+M} \to \mathcal{Y}^{s+M}\}, \mathcal{H}_{\widetilde{\mathbf{Y}}} \subset \{f : \mathcal{Y}^s \to \mathcal{Y}^s\}$ are RKHSs with associated kernels $k_{\mathbf{Y}}, k_{\widetilde{\mathbf{Y}}}$, respectively.*

We call $\nabla_F \mathcal{L}_{\mathcal{D}_s, \mathbf{X}}(Q_{[T]})\big|_{F=0}$ the Stein functional variational gradient operator. It inherits its name from SVGD, which internally is used to find the functional derivative of the $D_{\mathrm{KL}}$ term. The key idea of SFVGD is that by updating every particle function $f \sim Q$ via functional gradient descent in the direction of $\nabla_F \mathcal{L}_{\mathcal{D}_s, \mathbf{X}}(Q_{[T]})\big|_{F=0}(f)$, we carry out a gradient step in the distribution space. This increases the current overall functional ELBO value we want to maximize by pulling $Q$ closer to $Q^*$ and consequently also closer to the true posterior stochastic process $P_{f|\mathcal{D}}$.

## 3.3 ALGORITHMS

Based on the particle functions $f_1, \dots, f_r$, we can find an estimator of Eq. 16

$$
\begin{aligned}
\widetilde{\nabla}_F \mathcal{L}_{\mathcal{D}_s, \mathbf{X}}(Q_{[T]})\big|_{F=0}(f) = {}& \frac{1}{r} \sum_{i=1}^{r} \left[ \nabla_{\mathbf{f}_i \widetilde{\mathbf{x}}} \ell(\mathcal{D}_s, \mathbf{f}_i^{\widetilde{\mathbf{X}}}) \delta_{\mathbf{f}_i \widetilde{\mathbf{x}}}(\mathbf{f}^{\widetilde{\mathbf{X}}}) \right] \cdot \left[ \delta_{\widetilde{\mathbf{X}}_1}(\cdot), \dots, \delta_{\widetilde{\mathbf{X}}_s}(\cdot) \right]^{\top} \\
&+ \frac{\lambda}{r} \sum_{i=1}^{r} \left[ \nabla_{\mathbf{f}_i \mathbf{x}} \log p_{P_{f\mathbf{X}}}(\mathbf{f}_i^{\mathbf{X}}) k_{\mathbf{Y}}(\mathbf{f}_i^{\mathbf{X}}, \mathbf{f}^{\mathbf{X}}) + \nabla_{\mathbf{f}_i \mathbf{x}} k_{\mathbf{Y}}(\mathbf{f}_i^{\mathbf{X}}, \mathbf{f}^{\mathbf{X}}) \right] \\
&\cdot \left[ \delta_{\mathbf{X}_1}(\cdot), \dots, \delta_{\mathbf{X}_{s+M}}(\cdot) \right]^{\top},
\end{aligned}
\tag{18}
$$

---

**Algorithm 1:** Stein Functional Variational Gradient Descent Step    `sfvgd_step`

---

**Hyperparameters:** Dataset $\mathcal{D}$, log likelihood $\ell$, prior SP $P_f$, number of measure points $M$, sampling distribution $C_{\mathcal{X}}$ over $\mathcal{X}$, regularization parameter $\lambda$

**Input:** Set of particle functions $\{f_i\}_{i=1}^r$ treated as multi-output function $f$

**Output:** Input locations to update $\mathbf{X}$, Stein functional variational gradient (of $f$ evaluated at $\mathbf{X}$) $\mathbf{\Delta_f x}$

$\mathbf{X}_M \sim C_{\mathcal{X}}; \mathcal{D}_s = (\widetilde{\mathbf{X}}, \widetilde{\mathbf{y}}) \subset \mathcal{D}$

$\mathbf{X} = \left[\widetilde{\mathbf{X}}, \mathbf{X}_M\right]$

**for** $j = 1, \ldots, r$ **do**

$\quad \mathbf{\Delta}_{j,\ell} = \frac{1}{r} \sum_{i=1}^{r} \left[ \nabla_{\mathbf{f_i} \widetilde{\mathbf{x}}} \ell(\mathcal{D}_s, \mathbf{f_i}^{\widetilde{\mathbf{X}}}) \delta_{\mathbf{f_i} \widetilde{\mathbf{x}}}(\mathbf{f_j}^{\widetilde{\mathbf{X}}}) \right] \cdot \left[ \delta_{\widetilde{\mathbf{X}}_1}(\cdot), \ldots, \delta_{\widetilde{\mathbf{X}}_s}(\cdot) \right]^{\top}$

$\quad \mathbf{\Delta}_{j,\mathrm{KL}} = \frac{1}{r} \sum_{i=1}^{r} \left[ \nabla_{\mathbf{f_i} \mathbf{x}} \log p_{P_{f\,\mathbf{X}}}(\mathbf{f_i}^{\mathbf{X}}) k_{\mathbf{Y}}(\mathbf{f_i}^{\mathbf{X}}, \mathbf{f_j}^{\mathbf{X}}) + \nabla_{\mathbf{f_i} \mathbf{x}} k_{\mathbf{Y}}(\mathbf{f_i}^{\mathbf{X}}, \mathbf{f_j}^{\mathbf{X}}) \right]$

$\qquad \cdot \left[ \delta_{\mathbf{X}_1}(\cdot), \ldots, \delta_{\mathbf{X}_{s+M}}(\cdot) \right]^{\top}$

**end**

$\mathbf{\Delta_f x} = (\mathbf{\Delta}_\ell + \lambda \cdot \mathbf{\Delta}_{\mathrm{KL}})(\mathbf{X})$

---

**Algorithm 2:** Stein Functional Variational Neural Network

---

**Hyperparameters:** Same as for `sfvgd_step`

**Input:** Variational posterior $g(\cdot)$, optimizer `opt`

**Output:** Variational posterior $g(\cdot)$, which approximates the target distribution

**while** $\phi$ *not converged* **do**

$\quad f_i = g(h_\phi(\mathbf{X}, \xi_i)), \xi_i \sim p(\xi), \quad i = 1, \ldots, r$

$\quad \mathbf{X}, \mathbf{\Delta_f x} = \text{sfvgd\_step}(f)$

$\quad \phi = \text{opt}(\phi, \mathbf{X}, \mathbf{\Delta_f x})$

**end**

---

where we introduce a regularization parameter $\lambda \in \mathbb{R}_{\geq 0}$. Furthermore, if we set $\lambda = 1$, the estimator becomes an unbiased estimator of Eq. (16). Since $\mathcal{L}_{\mathcal{D},\mathbf{X}}$ is a lower bound of the log marginal likelihood $\log p(\mathcal{D})$, it would be preferable to update the particle functions via $\widetilde{\nabla}_F \mathcal{L}_{\mathcal{D},\mathbf{X}}(Q_{[T]})\big|_{F=0}$. However, the major computation bottleneck in Eq. 18 is the calculation of the score gradient $\nabla_{\mathbf{f_i} \mathbf{x}} \log p_{P_{f\,\mathbf{X}}}(\mathbf{f_i}^{\mathbf{X}})$ for all particle functions $f_i$, $i = 1, \ldots, r$ evaluated at $\mathbf{X}$. For example, if $P_f$ is a GP, then the costs of computing $\nabla_{\mathbf{f_i} \mathbf{x}} \log p_{P_{f\,\mathbf{X}}}(\mathbf{f_i}^{\mathbf{X}})$ are $\mathcal{O}((s+M)^3 r)$. In addition, the computation of all kernel values $k_{\mathbf{Y}}(\mathbf{f_i}^{\mathbf{X}}, \mathbf{f_j}^{\mathbf{X}})$, $i = 1, \ldots, r$, $j = 1, \ldots, r$ required in Eq. 18 costs $\mathcal{O}(r^2)$. However, this is usually small compared to the cost of computing the score gradient for the functional prior. We choose for $M$ a small constant number, since $\mathcal{L}_{\mathcal{D},[\mathbf{X}_\mathcal{D}, \mathbf{X}_M]}$ is a lower bound for the log marginal likelihood $\log p(\mathcal{D})$ for $M > 1$, and we set $r$ to a number of particle functions that can represent the posterior SP reasonably well. Thus, we are interested in estimating $\widetilde{\nabla}_F \mathcal{L}_{\mathcal{D},\mathbf{X}}(Q_{[T]})\big|_{F=0}$ with mini-batches. In principle, an unbiased estimate of $\ell(\mathcal{D}, \mathbf{f_i}^{\mathbf{X}_\mathcal{D}})$ is $n/s \cdot \ell(\mathcal{D}_s, \mathbf{f_i}^{\widetilde{\mathbf{X}}})$, which suggests that $\lambda = s/n$. Although (in general) $\mathcal{L}_{\mathcal{D}_s,\mathbf{X}}$ is not a lower bound of $\log p(\mathcal{D})$, we found in a practice setting that $\lambda$ to $s/n$ still results in reasonable performance. However, our theoretical framework gives the reassuring guarantee that if we use full-batch training, we would, in fact, maximize a lower bound of $\log p(\mathcal{D})$. In the following, we present two algorithms, namely Stein functional variational NNs and Stein functional variational gradient boosting (A.3.1), based on the estimated Stein functional variational gradient – i.e., they depend on the score gradient of the functional prior evaluated at $\mathbf{X}$. If there exists no analytical score gradient, we can use a score gradient estimator, as suggested in Sun et al. (2019). This only requires function samples of the prior process evaluated at $\mathbf{X}$, but estimating the score gradient is usually computationally expensive (Zhou et al., 2020). Since our approach builds upon SVGD, there exists an additional approach in our framework based on a gradient-free SVGD (Han & Liu, 2018) that only requires the evaluation of the marginal densities of the prior process.

**Stein Functional Variational Neural Network (SFVNN)** Sun et al. (2019) proposed to train neural networks (NNs) acting as function generators with the negative functional ELBO as loss, which they call Bayesian Functional Variational Neural Networks (BFVNNs). Such a function generator can be modeled via an NN with stochastic weights, which can be represented as a differentiable function $g: \mathcal{Z} \to \mathcal{Y}, z \mapsto g(z)$, where $z \in \mathcal{Z}$ consists of the deterministic input $\mathbf{x}$ and stochastic

inputs, i.e., we can model $z$ as a random variable $z \sim p(z|\mathbf{x})$. These NNs are applicable as long as the reparameterization trick (Kingma & Welling, 2014) can be used, i.e., there exists a random variable $\xi \in \Xi$ with $\xi \sim p(\xi)$ and a differentiable function $h_\phi : \mathcal{X} \times \Xi \to \mathcal{Z}$ parametrized by $\phi$ such that $h_\phi(\mathbf{x}, \xi) \sim p(z|\mathbf{x})$. With this, we can sample a function by sampling $\xi \sim p(\xi)$ and defining $f_\xi : \mathcal{X} \to \mathcal{Y}, \mathbf{x} \mapsto g(h_\phi(\mathbf{x}, \xi))$. In this case, we can write the gradient of Eq. 14 w.r.t. $\phi$ as

$$\nabla_\phi \mathcal{L}_{\mathcal{D}_s, \mathbf{x}}(Q_{[T]}) = \mathbb{E}_{\xi \sim p(\xi)} \left[ \nabla_{\mathbf{f}_\xi \tilde{\mathbf{x}}} \ell(\mathcal{D}_s, \mathbf{f}_\xi^{\tilde{\mathbf{X}}}) \nabla_\phi \mathbf{f}_\xi^{\tilde{\mathbf{X}}} \right]$$
$$- \mathbb{E}_{\xi \sim p(\xi)} \left[ \left( \nabla_{\mathbf{f}_\xi \mathbf{x}} \log p_{Q_\mathbf{x}}(\mathbf{f}_\xi^{\mathbf{X}}) - \nabla_{\mathbf{f}_\xi \mathbf{x}} \log p_{P_{f_\mathbf{x}}}(\mathbf{f}_\xi^{\mathbf{X}}) \right) \nabla_\phi \mathbf{f}_\xi^{\mathbf{X}} \right].$$

This is also the result obtained in Sun et al. (2019), where they then use a score estimator (namely, the spectral stein gradient estimator (SSGE; Shi et al., 2018)) to approximate $\nabla_\mathbf{y} \log p_{Q_\mathbf{x}}(\mathbf{f}_\xi^{\mathbf{X}})$. SSGE estimates the score gradient in an RKHS, i.e., the entropy gradient $\nabla_{\mathbf{f}_\xi \mathbf{x}} \log p_{Q_\mathbf{x}}(\mathbf{f}_\xi^{\mathbf{X}})$. Hence, the entropy gradient and the cross entropy gradient $\nabla_{\mathbf{f}_\xi \mathbf{x}} \log p_{P_{f_\mathbf{x}}}(\mathbf{f}_\xi^{\mathbf{X}})$ are taken in different functional spaces. Our SFVNN is based on the parameter gradient

$$\nabla_\phi \mathcal{L}_{\mathcal{D}_s, \mathbf{x}}(Q_{[T]}) = \mathbb{E}_{\xi \sim p(\xi)} \left[ \nabla_F \mathcal{L}_{\mathcal{D}_s, \mathbf{x}}(Q_{[T]})\big|_{F=0}(f_\xi)(\mathbf{X}) \cdot \nabla_\phi \mathbf{f}_\xi^{\mathbf{X}} \right] - \underbrace{\mathbb{E}_{\mathbf{f}^\mathbf{x} \sim Q_\mathbf{x}} \nabla_\phi \log p_{Q_\mathbf{x}, \phi}(\mathbf{f}^\mathbf{X})}_{=0},$$

where we use the general Stein functional variational gradient from Eq. 16. In contrast to Sun et al. (2019), we thereby directly take the functional gradient of the $D_{\mathrm{KL}}$ term in an RKHS, and our score gradients of the prior process are also subject to the implicit kernel smoothing.

**Runtime comparison between SVFNN and FVBNN** While FVBNN scales as $\mathcal{O}(r^3 + r^2(s + M))$ (because of SSGE), our approach scales only quadratically in $r$ (because of SVGD), allowing for a larger number of sample functions (see Appendix B).

## 3.4 ILLUSTRATIVE EXAMPLE

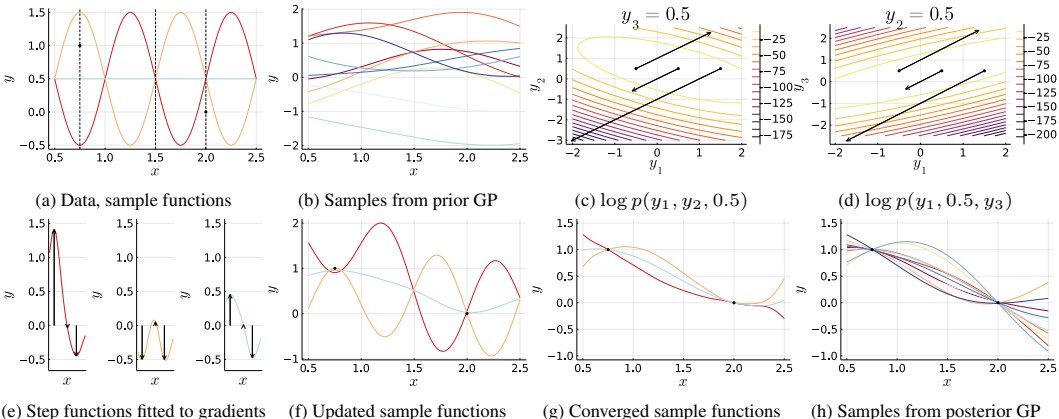

(a) Data, sample functions   (b) Samples from prior GP   (c) $\log p(y_1, y_2, 0.5)$   (d) $\log p(y_1, 0.5, y_3)$

(e) Step functions fitted to gradients   (f) Updated sample functions   (g) Converged sample functions   (h) Samples from posterior GP

Figure 1: Illustrative example of SFVGB. The points in (a) represent the given data points, and the dashed lines represent all $x$ values that define the marginal prior density in (c,d) w.r.t. the prior GP shown in (b). The arrows in (c) show the resulting SFVGD gradients.

Given three sample functions and two data points $\{(0.75, 1.0), (2.0, 0.0)\}$ (1a), we want to approximate the posterior GP (Figure 1h) w.r.t. the prior GP shown in Figure 1b and a Gaussian likelihood via SFVGB. Hence, we also sample a necessary measure point $x_M = 1.5$. The resulting three-dimensional marginal density is defined by the prior GP. SVGD gives us the optimal update direction for the sample function values to fit this marginal density (Figures 1c, 1d). We fit a kernel ridge regression to these directions after adding the $\log$ likelihood gradients at the two data points (Figure 1e). The resulting updated function samples after this SFVGD step can be seen in Figure 1f and the converged function samples in Figure 1g. Qualitatively comparing these converged sample functions in Figure 1g with sample functions from the exact posterior GP in Figure 1h reassures that we are are able to approximate the posterior GP in this toy example reasonably well.

Table 1: Comparison of different methods (columns) on small benchmark data sets (rows) using the average NLL (smaller is better) and RMSE over 10 train-test data splits with standard deviation in brackets. The best performing method for each data set is highlighted in bold.

| | Test negative log-likelihood | | | | Test root-mean-square error | | | |
|---|---|---|---|---|---|---|---|---|
| | SFVNN | FVBNN | BNN | GP | SFVNN | FVBNN | BNN | GP |
| Airfoil | **2.10** (0.17) | 2.29 (0.04) | 2.62 (0.12) | 2.50 (0.14) | **1.82** (0.20) | 1.97 (0.19) | 3.40 (0.40) | 2.77 (0.25) |
| Concrete | **2.99** (0.19) | 3.07 (0.05) | 3.25 (0.04) | 3.06 (0.05) | **4.58** (0.34) | 4.64 (0.54) | 6.18 (0.34) | 5.13 (0.40) |
| Diabetes | 5.42 (0.08) | 5.49 (0.03) | **5.41** (0.04) | 6.19 (0.38) | 54.57 (3.74) | 57.1 (2.48) | **52.7** (2.88) | 57.45 (6.6) |
| Energy | **0.62** (0.10) | 0.70 (0.09) | 2.26 (0.32) | 2.38 (0.05) | 0.44 (0.05) | **0.43** (0.08) | 2.37 (0.65) | 2.34 (0.23) |
| ForestF | 2.38 (0.44) | 1.84 (0.05) | **1.83** (0.05) | 4.65 (0.45) | 1.76 (0.31) | **1.51** (0.07) | **1.51** (0.08) | 1.56 (0.08) |
| Wine | 1.96 (1.45) | 1.47 (1.07) | -0.03 (0.07) | **-0.04** (0.06) | **0.11** (0.02) | 0.14 (0.02) | 0.21 (0.03) | 0.16 (0.03) |
| Yacht | **1.06** (0.30) | 1.11 (0.24) | 1.35 (0.19) | 2.86 (0.15) | 0.67 (0.26) | **0.61** (0.25) | 0.96 (0.28) | 3.95 (1.03) |
| Mean rank | 1.86 | 2.43 | 2.57 | 3.14 | 1.86 | 1.79 | 3.07 | 3.29 |

Table 2: Comparison of different methods (columns) on large benchmark data sets (rows) using the average NLL (smaller is better) and RMSE over 10 train-test data splits with standard deviation in brackets. The best performing method for each data set is highlighted in bold.

| | Test negative log-likelihood | | | Test root-mean-square error | | |
|---|---|---|---|---|---|---|
| | SFVNN | FVBNN | BNN | SFVNN | FVBNN | BNN |
| GPU | **4.73** (0.04) | 4.80 (0.03) | **4.73** (0.02) | 27.67 (1.26) | 29.6 (0.9) | **27.5** (0.67) |
| NavalT | **-6.91** (0.06) | -6.85 (0.08) | -5.03 (0.24) | **1.70E-4** (2.5E-5) | 1.94E-4 (3.3E-5) | 6.50E-4 (6.5E-5) |
| NavalC | **-6.53** (0.01) | -6.41 (0.05) | -6.44 (0.11) | **1.35E-4** (1.1E-4) | 2.39E-4 (3.5E-5) | 2.15E-4 (3.9E-5) |
| Protein | **2.85** (0.01) | 2.87 (0.01) | 2.96 (0.01) | **4.19** (0.04) | 4.27 (0.04) | 4.65 (0.05) |
| VideoMem | 11.39 (0.39) | **11.3** (0.10) | 11.4 (0.03) | 21119 (4910) | **20800** (2320) | 21800 (788) |
| VideoTime | **2.29** (0.05) | 2.53 (0.36) | 2.86 (1.02) | **2.51** (0.16) | 3.14 (0.88) | 3.83 (2.08) |
| Mean rank | 1.25 | 2.17 | 2.58 | 1.33 | 2.17 | 2.50 |

## 4 BENCHMARK STUDY

We further investigate the competitiveness of our approach using its neural network variant (SFVNN) with its closest neighbor, the functional variational Bayesian neural network (FVBNN) from Sun et al. (2019). We also include one well-established BNN baseline (Blundell et al., 2015) and the standard Gaussian Process for the small data sets (where analytical computation is feasible). For results from SVFGB we refer to Section A.3.2 in the Appendix. For most of the datasets, however, the NN provides a better fit to the data. Further details and a contextual bandits experiment can be found in Appendix A.5.

**Data and experimental setup** All data sets are standardized prior to model fitting and split into 90% training data and 10% test data. For the comparisons, this splitting process is repeated 10 times based on 10 different splits to also evaluate the variability of each method. Further details on data sets, data set-specific pre-processing, and their references can be found in the Appendix.

**Details on methods and comparisons** In order to provide a fair comparison between methods, we reproduce the best results reported by Sun et al. (2019) for FVBNN and BNN, and also use the same hyperparameters for our method except that while Sun et al. (2019) use $\lambda = 1$, we set $\lambda$ to $s/n$. Details for each procedure are given in the Appendix. We compare methods based on the negative log-likelihood (NLL) and the root mean squared error (RMSE) on each test data set and calculate the mean and standard deviation across all 10 data splits.

**Results** Results are summarized in Tables 1 and 2, indicating that SFVNN is competitive with other existing approaches for both small and large data sets. As the two functional approaches (SFVNN, FVBNN) optimize the same objective, we would expect them to perform similarly, which is confirmed by the results. We further observe that a weight space approach (the BNN) seems to work better than the functional approaches for a few datasets (in particular, for the Wine dataset, where this is expected as the outcome is of discrete nature).

## 5 CONCLUSION

We introduced a novel gradient descent in distribution space that allows us to update a set of particle functions in a general manner to resemble sample functions from a target process. SFVNN was found to be competitive with or to outperform FBVNN while having less computational costs.

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

# A APPENDIX

## A.1 PROOFS

### A.1.1 FUNCTIONAL DERIVATIVE OF THE RISK FUNCTIONAL

Assuming that $p(\mathbf{x}, y)L_{(\mathbf{x},y)}[f] \in L^1$ using Fubini's theorem, we find that

$$\mathcal{R}[f] = \mathbb{E}_{(\mathbf{x},y)\sim\mathbb{P}_{\mathbf{x}y}}L_{(\mathbf{x},y)}[f] = \int_{\mathcal{X}\times\mathcal{Y}} p(\mathbf{x}, y)L_{(\mathbf{x},y)}[f]d\mathbf{x}\,dy \tag{19}$$

$$= \int_{\mathcal{X}} p(\mathbf{x}) \underbrace{\int_{\mathcal{Y}} p(y|\mathbf{x})L_{(\mathbf{x},y)}[f]dy}_{=:\mathcal{L}(\mathbf{x},f(\mathbf{x}))}\,d\mathbf{x}. \tag{20}$$

Assuming that $\mathcal{L}$ is sufficiently smooth using the Euler-Lagrange derivative, we find that

$$\nabla_f\mathcal{R}[f](\mathbf{x}) = \frac{\partial\left(\mathcal{L}(\mathbf{x}, f(\mathbf{x}))\right)}{\partial f(\mathbf{x})} \tag{21}$$

$$= p(\mathbf{x})\int_{\mathcal{Y}} p(y|\mathbf{x})\frac{\partial L_{(\mathbf{x},y)}[f]}{\partial f(\mathbf{x})}dy \tag{22}$$

$$= p(\mathbf{x})\cdot\mathbb{E}_{y\sim\mathbb{P}_{y|\mathbf{x}}}\frac{\partial L_{(\mathbf{x},y)}[f]}{\partial f(\mathbf{x})}. \tag{23}$$

### A.1.2 FUNCTIONAL DERIVATIVE OF THE FUNCTIONAL ELBO

Let $F$ be an operator that depends on evaluation only with associated maps $F_{\widetilde{\mathbf{X}}} : \mathcal{Y}^s \to \mathcal{Y}^s, F_{\mathbf{X}} : \mathcal{Y}^{s+M} \to \mathcal{Y}^{s+M}$. Further, let $\mathcal{F}[F] = \mathcal{L}_{\mathcal{D}_s,\mathbf{x}}(Q_{[T]}) = \underbrace{\mathbb{E}_{\widetilde{\mathbf{y}}\sim Q_{[T]\widetilde{\mathbf{X}}}}\ell(\mathcal{D}_s, \widetilde{\mathbf{y}})}_{=\mathcal{F}_1[F_{\widetilde{\mathbf{X}}}]} - \underbrace{D_{\mathrm{KL}}(Q_{[T]\mathbf{X}}\|P_{f\mathbf{X}})}_{=\mathcal{F}_2[F_{\mathbf{X}}]}$.

In general, and under the assumption that $\ell$ is sufficiently smooth, we find that

$$\nabla_{F_{\widetilde{\mathbf{X}}}}\mathcal{F}_1[F_{\widetilde{\mathbf{X}}}] = \nabla_{F_{\widetilde{\mathbf{X}}}}\mathbb{E}_{\widetilde{\mathbf{y}}\sim Q_{\widetilde{\mathbf{X}}}}\ell(\mathcal{D}_s, \widetilde{\mathbf{y}} + F_{\widetilde{\mathbf{X}}}(\widetilde{\mathbf{y}})) = \mathbb{E}_{\widetilde{\mathbf{y}}\sim Q_{\widetilde{\mathbf{X}}}}\left[\nabla_{\widetilde{\mathbf{y}}}\ell(\mathcal{D}_s, \widetilde{\mathbf{y}} + F_{\widetilde{\mathbf{X}}}(\widetilde{\mathbf{y}}))\cdot\delta_{\widetilde{\mathbf{y}}}(\cdot)\right]. \tag{24}$$

Under the assumption that $\mathcal{H}_{\widetilde{\mathbf{Y}}} \subset \{f : \mathcal{Y}^s \to \mathcal{Y}^s\}$ is an RKHS with associated kernels $k_{\widetilde{\mathbf{Y}}}$, we find that

$$\mathcal{F}_1[F_{\widetilde{\mathbf{X}}} + \varepsilon G_{\widetilde{\mathbf{X}}}] - \mathcal{F}_1[F_{\widetilde{\mathbf{X}}}] = \mathbb{E}_{\widetilde{\mathbf{y}}\sim Q_{\widetilde{\mathbf{X}}}}\left[\ell(\mathcal{D}_s, \widetilde{\mathbf{y}} + F_{\widetilde{\mathbf{X}}}(\widetilde{\mathbf{y}}) + \varepsilon G_{\widetilde{\mathbf{X}}}(\widetilde{\mathbf{y}})) - \ell(\mathcal{D}_s, \widetilde{\mathbf{y}} + F_{\widetilde{\mathbf{X}}}(\widetilde{\mathbf{y}}))\right] \tag{25}$$

$$= \varepsilon\,\mathbb{E}_{\widetilde{\mathbf{y}}\sim Q_{\widetilde{\mathbf{X}}}}\left[\nabla_{\widetilde{\mathbf{y}}}\ell(\mathcal{D}_s, \widetilde{\mathbf{y}} + F_{\widetilde{\mathbf{X}}}(\widetilde{\mathbf{y}}))\cdot G_{\widetilde{\mathbf{X}}}(\widetilde{\mathbf{y}})\right] + \mathcal{O}(\varepsilon^2) \tag{26}$$

$$= \varepsilon\,\mathbb{E}_{\widetilde{\mathbf{y}}\sim Q_{\widetilde{\mathbf{X}}}}\left[\nabla_{\widetilde{\mathbf{y}}}\ell(\mathcal{D}_s, \widetilde{\mathbf{y}} + F_{\widetilde{\mathbf{X}}}(\widetilde{\mathbf{y}}))\cdot\langle k_{\widetilde{\mathbf{Y}}}(\widetilde{\mathbf{y}}, \cdot), G_{\widetilde{\mathbf{X}}}\rangle\right] + \mathcal{O}(\varepsilon^2) \tag{27}$$

$$= \varepsilon\,\langle\mathbb{E}_{\widetilde{\mathbf{y}}\sim Q_{\widetilde{\mathbf{X}}}}\left[\nabla_{\widetilde{\mathbf{y}}}\ell(\mathcal{D}_s, \widetilde{\mathbf{y}} + F_{\widetilde{\mathbf{X}}}(\widetilde{\mathbf{y}}))\cdot k_{\widetilde{\mathbf{Y}}}(\widetilde{\mathbf{y}}, \cdot)\right], G_{\widetilde{\mathbf{X}}}\rangle + \mathcal{O}(\varepsilon^2), \tag{28}$$

from which it follows that

$$\nabla_{F_{\widetilde{\mathbf{X}}}}\mathcal{F}_1[F_{\widetilde{\mathbf{X}}}]\big|_{F_{\widetilde{\mathbf{X}}}=0} = \mathbb{E}_{\widetilde{\mathbf{y}}\sim Q_{\widetilde{\mathbf{X}}}}\left[\nabla_{\widetilde{\mathbf{y}}}\ell(\mathcal{D}_s, \widetilde{\mathbf{y}} + F_{\widetilde{\mathbf{X}}}(\widetilde{\mathbf{y}}))\cdot k_{\widetilde{\mathbf{Y}}}(\widetilde{\mathbf{y}}, \cdot)\right]\big|_{F_{\widetilde{\mathbf{X}}}=0} \tag{29}$$

$$= \mathbb{E}_{\widetilde{\mathbf{y}}\sim Q_{\widetilde{\mathbf{X}}}}\left[\nabla_{\widetilde{\mathbf{y}}}\ell(\mathcal{D}_s, \widetilde{\mathbf{y}})\cdot k_{\widetilde{\mathbf{Y}}}(\widetilde{\mathbf{y}}, \cdot)\right]. \tag{30}$$

Under the assumption that $\mathcal{H}_{\mathbf{Y}} \subset \{f : \mathcal{Y}^{s+M} \to \mathcal{Y}^{s+M}\}$ is an RKHS with associated kernels $k_{\mathbf{Y}}$, it has been shown in Liu & Wang (2016) that

$$\nabla_{F_{\mathbf{x}}}\mathcal{F}_2[F_{\mathbf{X}}]\big|_{F_{\mathbf{X}}=0} = -\mathbb{E}_{\mathbf{y}\sim Q_{\mathbf{X}}}\left[\nabla_{\mathbf{y}}\log p_{P_{f\mathbf{X}}}(\mathbf{y})k_{\mathbf{Y}}(\mathbf{y}, \cdot) + \nabla_{\mathbf{y}}k_{\mathbf{Y}}(\mathbf{y}, \cdot)\right]. \tag{31}$$

Using the chain rule, we obtain

$$\nabla_F\mathcal{L}_{\mathcal{D}_s,\mathbf{x}}(Q_{[T]})\big|_{F=0}(f) = \nabla_{F_{\widetilde{\mathbf{X}}}}\mathcal{F}_1[F_{\widetilde{\mathbf{X}}}](\mathbf{f}^{\widetilde{\mathbf{X}}})\nabla_F\left[F_{\widetilde{\mathbf{X}}}\right](f)\big|_{F=0} \tag{32}$$

$$- \nabla_{F_{\mathbf{X}}}\mathcal{F}_2[F_{\mathbf{X}}](\mathbf{f}^{\mathbf{X}})\nabla_F\left[F_{\mathbf{X}}\right](f)\big|_{F=0}. \tag{33}$$

$$\tag{34}$$

---

**Algorithm 3:** Stein Functional Variational Gradient Boosting

---

**Hyperparameters:** Same as for `sfvgd_step`
**Input:** number of iterations $t_{\max}$, learning rate $\eta^{[t]}$ in the $t$-th iteration, set of initial particle functions
$\quad \{f_i^{[0]}\}_{i=1}^r$ treated as multi-output function $f^{[0]}$, multi-output base learner $\mathcal{I}_b$
**Output:** Set of particle functions $\{f_i^{[t_{\max}]}\}_{i=1}^r$, which approximate the target distribution
**for** $t = 0, \ldots, t_{\max} - 1$ **do**
$\quad\quad \mathbf{X}, \boldsymbol{\Delta}_{\mathbf{f}}\mathbf{x} = \texttt{sfvgd\_step}(f^{[t]})$
$\quad\quad f^{[t+1]} = f^{[t]} + \eta^{[t]} \cdot \mathcal{I}_b(\mathbf{X}, \boldsymbol{\Delta}_{\mathbf{f}}\mathbf{x})$
**end**

---

Since $F$ is evaluation-only dependent, we observe that $\nabla_F \left[ F_{\{\mathbf{x}\}} \right](f) = \delta_{\mathbf{x}}(\cdot)$ and conclude that

$$\nabla_F \mathcal{L}_{\mathcal{D}_s, \mathbf{x}}(Q_{[T]})\big|_{F=0}(f) = \nabla_{F_{\widetilde{\mathbf{X}}}} \mathcal{F}_1[F_{\widetilde{\mathbf{X}}}](\mathbf{f}^{\widetilde{\mathbf{X}}})\Big|_{F_{\widetilde{\mathbf{X}}}=0} \cdot \left[ \delta_{\widetilde{\mathbf{X}}_1}(\cdot), \ldots, \delta_{\widetilde{\mathbf{X}}_s}(\cdot) \right]^\top \tag{35}$$

$$- \nabla_{F_{\mathbf{X}}} \mathcal{F}_2[F_{\mathbf{X}}](\mathbf{f}^{\mathbf{X}})\Big|_{F_{\mathbf{X}}=0} \cdot \left[ \delta_{\mathbf{X}_1}(\cdot), \ldots, \delta_{\mathbf{X}_{s+M}}(\cdot) \right]^\top. \tag{36}$$

If $\mathcal{H}$ is assumed to be an RKHS with associated kernel $k_{\mathcal{X}}(\mathbf{x}, \cdot)$ then this becomes

$$\nabla_F \mathcal{L}_{\mathcal{D}_s, \mathbf{x}}(Q_{[T]})\big|_{F=0}(f) = \nabla_{F_{\widetilde{\mathbf{X}}}} \mathcal{F}_1[F_{\widetilde{\mathbf{X}}}](\mathbf{f}^{\widetilde{\mathbf{X}}})\Big|_{F_{\widetilde{\mathbf{X}}}=0} \cdot \left[ k_{\mathcal{X}}(\widetilde{\mathbf{X}}_1, \cdot), \ldots, k_{\mathcal{X}}(\widetilde{\mathbf{X}}_s, \cdot) \right]^\top \tag{37}$$

$$- \nabla_{F_{\mathbf{X}}} \mathcal{F}_2[F_{\mathbf{X}}](\mathbf{f}^{\mathbf{X}})\Big|_{F_{\mathbf{X}}=0} \cdot \left[ k_{\mathcal{X}}(\mathbf{X}_1, \cdot), \ldots, k_{\mathcal{X}}(\mathbf{X}_{s+M}, \cdot) \right]^\top. \tag{38}$$

## A.2 GRADIENT BOOSTING

GB (Friedman, 2001) is a powerful supervised learning algorithm where, iteratively, the residuals are minimized via so-called weak learners. The resulting model consists of a weighted ensemble of these weak learners. In its original form, tree-based learners were used as weak learners, which proved to be effective, especially in the presence of heterogeneous features. XGBoost (Chen & Guestrin, 2016) is a highly efficient algorithm that builds upon the GB paradigm, which proves to be a strong baseline for many structured, supervised regression and classification tasks.

## A.3 SFVGB

### A.3.1 SFVGB ALGORITHM

We treat the $r$ particle functions mapping from $\mathcal{X}$ to $\mathcal{Y}$ as a single function $f^{[0]}$ mapping from $\mathcal{X}$ to $\mathcal{Y}^r$ – i.e., we identify the $i$-th sample function $f_i^{[0]}, i = 1, \ldots, r$ with the $i$-th component of $f^{[0]}$. Analogously to standard GB, we choose a base learner $\mathcal{I}_b$ which defines a hypothesis space $\mathcal{H} \subset \{f : \mathcal{X} \to \mathcal{Y}^r\}$. While this requires the base learner $\mathcal{I}_b$ to be a multi-output learner, it is always possible to use an ensemble of single-output learners. With this, SFVGB is vanilla GB of a multi-output function $f$ where the loss is the negative functional ELBO. Consequently, we update $f^{[t]}$ in the $t$-th iteration via

$$f^{[t+1]} = f^{[t]} + \eta^{[t]} \mathcal{I}_b\left( \mathbf{X}, \left[ \widetilde{\nabla}_F \mathcal{L}_{\mathcal{D}_s, \mathbf{x}}(Q_{[T]})\big|_{F=0}(f_1^{[t]})(\mathbf{X}), \ldots, \widetilde{\nabla}_F \mathcal{L}_{\mathcal{D}_s, \mathbf{x}}(Q_{[T]})\big|_{F=0}(f_r^{[t]})(\mathbf{X}) \right]^\top \right),$$

using Eq. 18 and a (potentially adaptive) learning rate $\eta^{[t]}$.

### A.3.2 SFVGB REGRESSION EXPERIMENTS

We also test SFVGB on the small regression datasets and compare it to 1) the approach proposed in Malinin et al. (2021), i.e., uncertainty quantification approaches by randomly subsampling the data in every iteration of a stochastic gradient boosting (SGB) model and 2) boosting generalized linear model (GLMB Buehlmann, 2006) as a baseline approach. As base learners, SGB uses trees and GLMB uses linear models. Our model uses a Nadaraya-Watson kernel regression variant, which does not scale the resulting sum of kernel functions. This is the natural learner if the hypothesis space $\mathcal{H}$ is assumed to be an RHKS, as discussed in section 2.1. The hyperparameters of the SFVGD step are the same as the ones we used for SFVNN. The results after 1000 iterations are summarized in Table 4.

Table 3: Comparison of boosting approaches on the small benchmark data sets (columns) using the average NLL (smaller is better) over 10 train-test data splits with standard deviation in brackets. The best performing method for each data set is highlighted in bold.

|       | Airfoil     | Concrete    | Diabetes    | Energy      | ForestF     | Wine        | Yacht       |
|-------|-------------|-------------|-------------|-------------|-------------|-------------|-------------|
| SFVGB | 2.87 (0.18) | 3.41 (0.13) | 7.04 (0.67) | 2.16 (0.10) | 6.31 (0.69) | 2.37 (2.00) | 3.21 (0.52) |
| GLMB  | 3.10 (0.03) | 3.93 (0.03) | 5.50 (0.03) | 3.28 (0.01) | 1.84 (0.07) | 0.72 (0.02) | 3.80 (0.04) |
| SGB   | 1.98 (0.05) | 3.06 (0.10) | 5.51 (0.07) | 0.79 (0.49) | 1.86 (0.08) | 0.11 (0.44) | 0.36 (0.23) |

Table 4: Comparison of boosting approaches on the small benchmark data sets (columns) using the average RMSE (smaller is better) over 10 train-test data splits with standard deviation in brackets. The best performing method for each data set is highlighted in bold.

|       | Airfoil     | Concrete    | Diabetes    | Energy      | ForestF     | Wine        | Yacht       |
|-------|-------------|-------------|-------------|-------------|-------------|-------------|-------------|
| SFVGB | 3.31 (0.26) | 6.71 (0.52) | 57.8 (0.67) | 1.93 (0.14) | 1.53 (0.07) | 0.18 (0.04) | 4.83 (1.24) |
| GLMB  | 4.85 (0.27) | 10.5 (1.03) | 53.3 (3.42) | 3.07 (0.21) | 1.51 (0.09) | 0.27 (0.04) | 8.51 (1.18) |
| SGB   | 2.06 (0.21) | 5.06 (0.53) | 57.6 (2.71) | 0.57 (0.09) | 1.52 (0.09) | 0.29 (0.14) | 0.85 (0.45) |

**Further experimental details** Model tuning of SGB is done as explained in Malinin et al. (2021). Here, different tree depths $\in \{3, 4, 5, 6\}$, learning rates $\{0.001, 0.01, 0.1\}$, and numbers of samples $\in \{0.25, 0.5, 0.75\}$ of approaches are trained on the first 80% of the training data and evaluated on the latter 20%. The GLMB approach is tuned using a 10-fold cross-validation to determine the number of stopping iterations, which is the only hyperparameter of the model.

**Results** Results show that the SFVGB can often improve over the GLMB baseline but still yields inferior performance on some data sets. This is, in particular, the case if there is a rather discrete outcome space (e.g., Wine which only consists of values 0, 1, and 2). The SGB model, in turn, works better than both the SFVGB and GLMB in most cases. This is likely due to the much more flexible base learner structure (as SGB like most of the state-of-the-art boosting approaches uses trees, whereas GLMB uses linear regression and ours uses kernel regression).

## A.4 NUMERICAL EXPERIMENTS: FURTHER DETAILS

### A.4.1 FURTHER DATA DETAILS

We use the benchmark data setup proposed by Hernandez-Lobato & Adams (2015) and Sun et al. (2019) for evaluating probabilistic regression approaches. This setup includes selected data sets from the UCI repository – namely, the four smaller data sets Concrete, Energy, Wine, Yacht, and the four larger data sets Naval, Protein, Video (Memory and Time), and GPU. In addition to Sun et al. (2019), we also compare our approaches on three additional smaller data sets (Airfoil, Diabetes, Forest Fire) and further investigate the second task on the Naval data set (i.e., we examine both the compressor decay and the turbine decay state coefficient, referred to as NavalC and NavalT, respectively). In Table 5, the data characteristics and pre-processing steps are listed.

### A.4.2 FURTHER EXPERIMENTAL DETAILS

BNN approaches are fitted using the recommended architecture and tuning parameters by Sun et al. (2019). We reduced the epochs from 10,000 to 1,000 epochs for the smaller data sets to reduce the computational runtime. This did not negatively impact the BNN's performance. In all our benchmark experiments, we follow the setup for BNNs and FVBNNs described by Sun et al. (2019).

## A.5 FURTHER RESULTS

Here, we provide further results using the application of probabilistic methods for contextual bandits.

**Contextual Bandits** One important application of uncertainty-aware models is for exploration, as in Bayesian optimization, reinforcement learning, or bandits. Following Sun et al. (2019), we evaluate SFVNN using the contextual bandits benchmark by Riquelme et al. (2018) by re-runing the settings investigated in Sun et al. (2019) and report the cumulative regret based on the best expected

Table 5: Data set characteristics, additional pre-processing and references.

| Dataset | # Obs. | # Feat. | Pre-processing | Reference |
|---------|--------|---------|----------------|-----------|
| Airfoil | 1503 | 5 | - | Dua & Graff (2017) |
| Concrete | 1030 | 8 | - | Yeh (1998) |
| Diabetes | 442 | 10 | - | Dua & Graff (2017) |
| Energy | 768 | 8 | - | Tsanas & Xifara (2012) |
| ForestF | 517 | 12 | logp1 transformation for `area`; numerical representation for `month` and `day` | Cortez & Morais (2007) |
| Wine | 178 | 13 | - | Dua & Graff (2017) |
| Yacht | 308 | 6 | - | Dua & Graff (2017) |
| GPU | 241600 | 14 | only use run 1 as outcome | Ballester-Ripoll et al. (2017) |
| NavalT | 11934 | 15 | drop features with zero variance | Coraddu et al. (2014) |
| NavalC | 11934 | 15 | drop features with zero variance | Coraddu et al. (2014) |
| Protein | 45730 | 9 | - | Dua & Graff (2017) |
| Video | 68784 | 19 | drop highly correlated features; drop `id` and `b_size`; use dummy-coding for `codec` and `o_codec` | Dua & Graff (2017) |

Table 6: Relative contextual bandits regret (relative to the cumulative regret of Uniform sampling) for different data sets (columns) and methods (rows). Numbers in brackets of methods indicate the network sizes. Reported numbers are the mean (and standard derivation in brackets) over 5 trials. The best algorithms per data set are highlighted in bold.

| | Adult | Census | Covertype | Jester | Mushroom | Statlog | Wheel |
|---|-------|--------|-----------|--------|----------|---------|-------|
| BNN (50) | 95.79 (1.33) | 66.12 (5.10) | 61.18 (1.91) | 83.70 (3.31) | 7.81 (9.39) | 29.2 (4.05) | 82.94 (22.2) |
| BNN (500) | 99.16 (1.47) | 94.38 (11.9) | 74.55 (7.32) | 79.08 (5.84) | 9.31 (11.1) | 64.5 (13.3) | **8.892** (15.0) |
| BootRMS | 82.74 (8.16) | 53.40 (18.0) | 36.95 (10.3) | 63.3 (10.74) | 7.11 (5.61) | 1.81 (2.70) | 119.16 (3.7) |
| Dropout | 85.46 (4.74) | 37.27 (12.3) | 39.55 (5.84) | 65.11 (9.62) | 4.43 (6.83) | 2.89 (4.26) | 28.61 (12.1) |
| FVBNN (50, 50, 50) | 72.84 (11.5) | 30.46 (28.4) | **24.29** (14.7) | 50.45 (20.0) | 2.92 (10.3) | 3.46 (3.75) | 13.62 (14.7) |
| FVBNN (50) | 75.05 (7.66) | 40.65 (14.9) | 46.57 (3.99) | 57.28 (17.8) | **2.39** (6.24) | **1.58** (2.09) | 24.24 (22.0) |
| ParamNoise | 89.00 (5.24) | 57.86 (15.5) | 48.18 (9.84) | 66.52 (13.1) | 6.74 (5.89) | 7.69 (3.57) | 21.93 (15.4) |
| SFVNN (50, 50, 50) | **71.99** (7.29) | 36.65 (30.3) | 28.47 (16.4) | **50.09** (21.2) | 7.81 (6.15) | 4.52 (4.12) | 44.12 (27.7) |
| SFVNN (50) | 79.62 (5.44) | **30.05** (13.1) | 29.24 (11.4) | 55.61 (18.0) | 6.88 (7.99) | 4.01 (2.19) | 91.95 (33.2) |
| Uniform Sampling | 100.0 (0.00) | 100.0 (0.00) | 100.0 (0.00) | 100.0 (0.00) | 100.0 (0.00) | 100.0 (0.00) | 100.0 (0.00) |

reward. Following Riquelme et al. (2018), we use the benchmark data sets Adult, Census, Covertype, Financial, Jester, Mushroom, Statlog, and Wheel. For these, the rewards are deterministic, and the regret is equal to the best realized reward. As in previous works, we report the regret as a relative value, relative to a random uniform sampling procedure that emulates Thompson Sampling (see Riquelme et al., 2018). As comparison methods, we use the BNN (with 50 and 500 units), three spinoffs of the NeuralLinear algorithm (namely, the Bootstrapped NN trained with RMSprop (BootRMS), Parameter Noise (ParamNoise), and Dropout (see Riquelme et al., 2018, for more details)), as well as two variants of the FVBNN (with one and three layers each with 50 units). Experiments are run 5 times with shuffled contexts, for which we report mean and standard deviation of the relative cumulative regret.

Results are given in Table 6, suggesting that both FVBNN and SFVBNN are well- and particularly similar-performing methods in the application of contextual bandits.

# B RUNTIME EXPERIMENT

## B.1 SETUP

We trained FVBNN and SVFNN 5 times on the Energy data set for each number of particle functions $r \in \{50, 100, 150\}$ and plotted the resulting runtimes in Figure 2. It becomes apparent that the number of particle functions influences the runtime of FVBNN more substantially than SFVNN, as we expect from our computational complexity analysis.

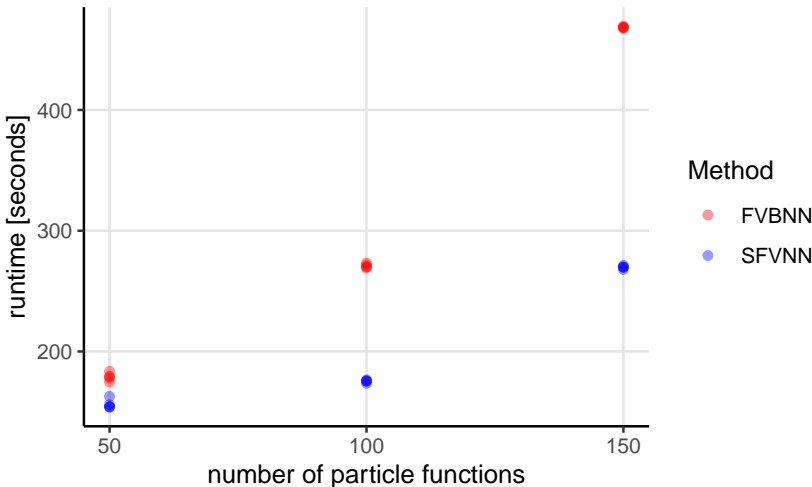

Figure 2: Comparision of the runtimes of FVBNN and SVFNN on the Energy data set with 5 repetitions for each number of particle functions

### B.2 COMPUTATIONAL ENVIRONMENT

All experiments and benchmarks were carried out on an internal cluster with Intel(R) Xeon(R) CPU E5-2650 v2 @ 2.60GHz, 32 cores, 64 GB Random-access memory, and operating system Ubuntu 20.04.1 LTS.

