# OpenReview forum: "Approximate Bayesian Inference with Stein Functional Variational Gradient Descent"
_ICLR.cc/2023/Conference — ICLR 2023 poster_

### Official Review · Reviewer_9cu5 · 2022-10-22

**Confidence:** 4
**Correctness:** 4
**Technical Novelty And Significance:** 2
**Empirical Novelty And Significance:** 2
**Recommendation:** 5

**Clarity, Quality, Novelty And Reproducibility:**

Quality: The paper presents an approach to generalize the Stein variational derivative. However, I did not see any motivations or analytical examples for the proposed methods. What is the quality of this method, at least in some simple examples?

**Strength And Weaknesses:**

Weakness: Please do not use blue and red colors on the paper.

**Summary Of The Paper:**

The authors study general variational algorithms, which generalize Stein variational gradient descent (SVGD) in functional space. They generalize the functional space in minimizing the KL divergence step. They apply the algorithms in Bayesian neural networks and ensemble gradient boosting. The numerical experiments demonstrate the efficiency of their algorithms.

**Summary Of The Review:**

This paper provides a method to approximate the Stein variational gradient in general settings. However, there is no analytical or motivation examples to support the proposed method.

---

> ### Author Response · Authors · 2022-11-15
> **Response and Revision**
>
> We thank the reviewer for the feedback and for taking the time to read our paper. Following your suggestions, we have removed the red and blue coloring of the text. Please further note that an illustrative example of the quality of our method can be found in the paper in Figure 1.
> We would be grateful for further comments on what the reviewer found unclear or where our approach is lacking novelty and significance so that we have the opportunity to improve our manuscript in this respect.

---

### Official Review · Reviewer_bTr3 · 2022-10-24

**Confidence:** 4
**Correctness:** 4
**Technical Novelty And Significance:** 3
**Empirical Novelty And Significance:** 2
**Recommendation:** 6

**Clarity, Quality, Novelty And Reproducibility:**

The description of the method is clear and it is of high theoretical quality. However, the empirical quality seems limited and the novelty is questionable, especially since many closely related works are not discussed. The experiments seem reproducible.

**Strength And Weaknesses:**

Strengths:
- The problem of function-space inference is well-motivated.
- The paper is clearly written.
- The theoretical derivations seem to be correct.

Weaknesses:
- The related work is incompletely acknowledged.
- The experiments seem rather inconclusive.

Comments:
- The proposed SFVNN seems (at least superficially) very similar to [1] and to some lesser degree to [2]. Other functional BNN approaches that seem worth mentioning are [3] and [4].
- The paper relies a lot on the KL divergence estimator from Sun et al. However, [5] has shown that the KL divergence is generally infinite between many different function-space distributions and that the estimator is thus ill-defined. Could the authors comment on that?
- In the presented regression experiments, the proposed method only outperforms the baselines on a few datasets. Could the authors comment on why it doesn't work well on the others?
- Could the authors comment on the runtime of the proposed method? I would naively think that it would probably be slower than the baselines.
- Since NNs are not necessarily much better on these regression tasks than, e.g., GPs, I think an image classification task would be a more suited experiment to motivate the method, similar to e.g. [1,2].
- Since [2] have shown that their method works better than the functional SVGD [1], it seems like that might also outperform the proposed method and should be a baseline in the experiment.
- The notation is sometimes hard to parse (up to 4 stacked subscripts), so maybe that could be made lighter.

[1] https://arxiv.org/abs/2106.10760

[2] https://arxiv.org/abs/2106.11642

[3] https://arxiv.org/abs/2008.08400

[4] https://hudsonchen.github.io/papers/Tractable_Function_Space_Variational_Inference_in_Bayesian_Neural_Networks.pdf

[5] https://arxiv.org/abs/2011.09421

**Summary Of The Paper:**

The paper proposes to perform Stein Variational Gradient Descent directly in the function space of neural networks or gradient-boosted models, thus offering two new methods. It derives theoretically how these approaches can be approximated in practice and shows empirically how they perform on simple benchmark regression tasks.

**Summary Of The Review:**

Overall, the method seems theoretically well-motivated, but the weak experiments and insufficient discussion of related work currently hinder me from recommending acceptance. If the authors could provide stronger experiments with relevant baselines on realistic data, I would be willing to change my assessment.

UPDATE: I have increased my score thanks to the changes made during the rebuttal.

---

> ### Author Response · Authors · 2022-11-15
> **Response and Revision**
>
> We thank the reviewer for the detailed comments and for taking the time to read our manuscript! In the following, we would like to address the 2 mentioned weaknesses first, and then further answer on the remaining specific points.
>
> #### Related work
>
> We appreciate the suggestions for improvement of our related literature section.  As now stated in our updated paper, both approaches by D'Angelo et al. ([1], [2]) focus on settings where the priors are given w.r.t. weights, and hence we do not see these approaches as closest neighbors. In [1] an approach similar to Wang is shown, which can incorporate functional priors but only if an analytical posterior process density exists. However, the authors do not sample measure points; hence, we can conclude from our theoretical framework that they are not maximizing a lower bound of the functional ELBO between the true posterior process and their overall approximation in this case. We have added the suggested related literature as well as a discussion in the revised version of our paper. We thank the reviewer as this further strengthens the presentation of our work as a more general and principled framework.
> We also thank the reviewer for bringing up the interesting idea of the combination of our approach and deep ensembles, i.e., how the inclusion of measure points affects the resulting deep ensemble. We think that this is worth investigating more comprehensively and will be mentioned in our revised paper as well.
>
> #### Experiments
>
> Since Sun et al. and our approach minimize the same objective function, both can be expected to perform similarly. However, the goal of our work was not to propose a method that outperforms Sun. Instead, we here derive a general gradient descent method that enables us to connect Sun's theoretical framework (and implied guarantees) with, e.g., current heuristic approaches in deep ensemble learning.
> This is also why we did not include a comparison with, e.g., [2], as suggested by the reviewer. While they showed that they outperform the approach proposed by Wang, this is not the functional BNN theory proposed by Sun et al. upon which our work builds.
> We further like to point out that in the course of deriving this general framework, we also created a method that only scales quadratically in the number of ensemble members, while the approach by Sun et al. scales cubically (because of the SSGE). This lower computational complexity results in a shorter runtime of the method compared to theirs. We discussed this already in the originally submitted version of the manuscript. Maybe the reviewer missed that part?
> We have now added a discussion to the results section of the paper, explaining the given results.
>
> #### Further comments
>
> While we did not encounter any problems with our objective in practice (regarding the infiniteness of the KLD when the variational family $\mathcal{Q}$ is parametric). One possible improvement is the grid functional KLD proposed by Ma and Hernández-Lobato which can be straightforwardly plugged into our framework and thereby potentially fix some of the shortcomings of the functional KL. We now mention this in our updated version and thank the reviewer for pointing this out. However, also note that SFVGD itself does not assume $\mathcal{Q}$ to be parametric.
>
> SFVNN can be seen as a scalable and more flexible (the likelihood and prior process do not have to be Gaussian) GP; hence, we see it as an essential benchmark to test its capabilities in regression with uncertainty.

---

> > ### Comment · Reviewer_bTr3 · 2022-11-15
> > **Thanks**
> >
> > Thank you for your responses. If indeed the SFVNN is meant to be rather an alternative to a GP than to a deep neural network, I would expect to see a GP as a baseline in the experiments. Moreover, I do appreciate that you report asymptotic runtime complexities, but in practice (on finite data) the runtime often depends more on the constants than on the asymptotics. I would therefore appreciate seeing some actual empirical wallclock runtime results on the experiments, so it would become obvious whether the theoretical runtime improvements can be achieved in practice.

---

> > > ### Author Response · Authors · 2022-11-16
> > > **Response and Revision**
> > >
> > > Thank you for further strengthing our paper; we included a GP as another baseline
> > >
> > > |           | **SFVNN (NLL)**   | **FVBNN (NLL)**   | **BNN (NLL)**      | **GP (NLL)**       | **SFVNN (RMSE)**     | **FVBNN (RMSE)**   | **BNN (RMSE)**     | **GP (RMSE)**      |
> > > |---------------|--------------|-------------|--------------|--------------|---------------|-------------|-------------|-------------|
> > > | **Airfoil**   | 2.10 (0.17)  | 2.29 (0.04) | 2.62 (0.12)  | 2.50 (0.14)  | 1.82 (0.20)   | 1.97 (0.19) | 3.40 (0.40) | 2.77 (0.25) |
> > > | **Concrete**  | 2.99  (0.19) | 3.07 (0.05) | 3.25 (0.04)  | 3.06 (0.05)  | 4.58 (0.34)   | 4.64 (0.54) | 6.18 (0.34) | 5.13 (0.40) |
> > > | **Diabetes**  | 5.42 (0.08)  | 5.49 (0.03) | 5.41 (0.04)  | 6.19 (0.38)  | 54.57 (3.74)  | 57.1 (2.48) | 52.7 (2.88) | 57.45 (6.6) |
> > > | **Energy**    | 0.62 (0.10)  | 0.70 (0.09) | 2.26 (0.32)  | 2.38 (0.05)  | 0.44 (0.05)   | 0.43 (0.08) | 2.37 (0.65) | 2.34 (0.23) |
> > > | **ForestF**   | 2.38 (0.44)  | 1.84 (0.05) | 1.83 (0.05)  | 4.65 (0.45)  | 1.76   (0.31) | 1.51 (0.07) | 1.51 (0.08) | 1.56 (0.08) |
> > > | **Wine**      | 1.96 (1.45)  | 1.47 (1.07) | -0.03 (0.07) | -0.04 (0.06) | 0.11 (0.02)   | 0.14 (0.02) | 0.21 (0.03) | 0.16 (0.03) |
> > > | **Yacht**     | 1.06 (0.30)  | 1.11 (0.24) | 1.35 (0.19)  | 2.86 (0.15)  | 0.67 (0.26)   | 0.61 (0.25) | 0.96 (0.28) | 3.95 (1.03) |
> > > | **Mean rank** | 1.86         | 2.43        | 2.57         | 3.14         | 1.86          | 1.79        | 3.07        | 3.29        |
> > >
> > > and added a runtime experiment to the appendix, which supports our claim.

---

> > > > ### Comment · Reviewer_bTr3 · 2022-11-27
> > > > **Thanks again**
> > > >
> > > > Thanks for the new experiments, I have updated my score.

---

### Official Review · Reviewer_qRMB · 2022-10-25

**Confidence:** 3
**Clarity, Quality, Novelty And Reproducibility:** See above for comments on clarity and…
**Correctness:** 3
**Technical Novelty And Significance:** 2
**Empirical Novelty And Significance:** 2
**Recommendation:** 5

**Strength And Weaknesses:**

### Strengths

This work attempts to provide a rigorous formulation of function-space, SVGD-like algorithms.  This is a sensible problem, as similar particle-based VI approaches have shown promises in the past, including for BNN inference where their function-space analogues have been studied in less rigorous ways.  A rigorous formulation would allow the principled use of such algorithms, and may also identify suboptimal constructions in past algorithms.

### Weaknesses

My main concerns are the following:

1. The current manuscript did not discuss previous work adequately.  It is not obvious from a first reading what efforts have been made to construct function-space SVGD algorithms: this work only cited Wang et al (2019), and neglected the works of D'Angelo et al (2021a; 2021b) which studied similar issues.  Moreover, there is no comparison between the update rules derived in this work and those in past works; and there is no empirical comparison either.  This leaves it very unclear how new derivations in this work have practical implications.

2. The presentation of the technical contents is quite confusing, which makes it difficult to understand or verify the claims.  Gradients were presented without specifying the inner product structure they are defined with.  The same gradient notation can be used to refer to both L2 and RKHS gradients (e.g., Eq. 3 and the discussion below).  Typos such as in Eq. 8 further complicated reading (Eq. 8 refers to the $\mathcal{H}$-Wasserstein gradient in a space of distribution over parameters, not "functions" which do not have to exist there).  It takes a lot of guesswork -- and familiarity with the background -- to understand what the authors really meant, and it should not have happened in a work that attempts to present a rigorous formulation for ideas that already exist (in part, and in various forms) in the past.

These two issues make it difficult to evaluate the manuscript in its present form, although I'm willing to go through it again once they are clarified.

3. A less important issue, which should nonetheless be made clear, is that all discussions in this work only applies to the full-batch training setting, where the sampled "measurement points" (Sun et al, 2019) always include the entire training set.  This is an inherent limitation (Burt et al, 2020) of the framework of Sun et al (2019), and the readers should be made aware about it.

### References

* Wang, Z., Ren, T., Zhu, J., & Zhang, B. (2018). Function Space Particle Optimization for Bayesian Neural Networks. In International Conference on Learning Representations.
* D'Angelo, Francesco, and Vincent Fortuin. "Repulsive deep ensembles are bayesian." Advances in Neural Information Processing Systems 34 (2021): 3451-3465.
* D'Angelo, Francesco, Vincent Fortuin, and Florian Wenzel. "On stein variational neural network ensembles." arXiv preprint arXiv:2106.10760 (2021).
* Burt, D. R., Ober, S. W., Garriga-Alonso, A., & van der Wilk, M. (2020). Understanding variational inference in function-space. arXiv preprint arXiv:2011.09421.

**Summary Of The Paper:**

This work studies function-space BNN inference, and formulated function-space variants of the Stein variational gradient descent algorithm.  The algorithm is evaluated on synthetic and tabular regression data.

**Summary Of The Review:**

While this work tackles an important problem and appears to have interesting contributions, the clarity issues make it difficult to judge its merits.

(My ratings below reflect the uncertainty due to the clarity issues, and I'm willing to update them after clarifications.)

---

> ### Author Response · Authors · 2022-11-15
> **Response and Revision**
>
> We thank the reviewer for the comments and for taking the time to suggest improvements of our manuscript. In the following, we answer the three main weakness points mentioned
>
> #### Related Work
>
> We appreciate the suggestions for improvement of our related literature section.  As now stated in our updated paper, both approaches by D'Angelo et al. (2021a, 2021b) focus on settings where the priors are given w.r.t. weights, and hence we do not see these approaches as closest neighbors. In 2021a an approach similar to Wang is shown, which can incorporate functional priors but only if an analytical posterior process density exists. However, the authors do not sample measure points; hence, we can conclude from our theoretical framework that they are not maximizing a lower bound of the functional ELBO between the true posterior process and their overall approximation in this case. We have added a discussion to our paper and also mention related literature regarding deep ensembles. We thank the reviewer as this further strengthens the presentation of our work as a more general and principled framework.
> We also thank the reviewer for bringing up the interesting idea of the combination of our approach and deep ensembles, i.e., how the inclusion of measure points affects the resulting deep ensemble. We think that this is worth investigating more comprehensively.
>
> #### Technical Contents
>
> We further clarified the relevant inner product structures. However, Eq. 8 is, in fact, a functional derivative: We consider the KLD between the current approximating density q transformed by the sum of the identity function and a function f, and the target density. Restricting the space of functions to an RKHS gives the desired result. The familiar reader recognizes that this is exactly Theorem 3.3 proven in "Stein Variational Gradient Descent: A General Purpose Bayesian Inference Algorithm" by Liu et al..
>
> #### Full-batch training
>
> On page 7, we state that when using batch sampling, we do not maximize a lower bound of the marginal log likelihood. We, however, agree with the reviewer that this should be stressed more and revised our manuscript accordingly.

---

### Official Review · Reviewer_NXDp · 2022-10-25

**Confidence:** 3
**Correctness:** 2
**Technical Novelty And Significance:** 3
**Empirical Novelty And Significance:** 2
**Recommendation:** 5

**Clarity, Quality, Novelty And Reproducibility:**

The paper is mostly well written, despite a few typos. The contribution seems novel. There's no mention of whether code would be released for reproducibility purposes.

Minor clarity issues:
* $\mathcal{I}_b$ could have an explicit definition.
* The integral in the denominator in Eq. 7 should be over $\boldsymbol{\theta}$, not $\mathbf{x}y$.
* A few parts of the first paragraph in Sec. 2.1 are missing the space between the period and the beginning of the next sentence.

**Strength And Weaknesses:**

### Strengths
* Compared to previous work (Sun et al. 2019; Wang et al., 2019), the proposed approach more naturally translates SVGD to function spaces by starting its derivation from functional gradients.
* Theoretical results are provided on the validity of the functional gradient estimators.
* Experimental results show performance improvements against previous function-space variational approach FVBNN.

### Weaknesses
* Related work on SVGD could mention some recent work on other forms of SVGD, such as second-order methods, matrix-valued kernels, and its convergence analysis.
* Related work on BNNs only mentions variational inference approaches, while Markov chain Monte Carlo methods, such as stochastic gradient Hamiltonian Monte Carlo, have also shown some success on inference for BNNs, though usually in problems of low dimensionality.
* Background on gradient boosting could be expanded.
* What is most concerning to me is that practical performance gains seem marginal when considering the results in Table 1.They are mostly within the +/- 1 std. deviation margin.
* There is no discussion on the results in Table 1. For example, why did the BNN (using Bayes by backprop) baseline perform significantly better than the functional-gradient methods in a few of the benchmarks? And importantly, what would be a few reasons for the performance of the proposed SFVNN and the FVBNN baseline to be so similar in most cases?
* In the appendix, there were no baselines for the comparisons with SFVGB.


**Summary Of The Paper:**

This paper proposes a method to perform approximate Bayesian inference in function spaces with an emphasis on applications to Bayesian neural networks and gradient boosting. The proposed approach extends Stein variational gradient descent (Liu & Wang, 2016) to function spaces by applying it to minimise the KL divergence with respect to a posterior stochastic process. The approach is made tractable by approximations propagating the functional gradients to the parameter space of the models being learnt. Experiments learning BNNs and gradient boosting models on benchmark datasets are presented showing performance improvements with respect to baselines.

**Summary Of The Review:**

The paper's contributions seem to be mostly on the formulation of a new method with theoretical support, but with only marginal practical performance gains in experimental evaluations, which cast doubts regarding its potential impact.

---

> ### Author Response · Authors · 2022-11-15
> **Response and Revision**
>
> We thank the reviewer for the comments and for taking the time to suggest improvements to our manuscript. In the following, we answer the two main weak points mentioned (related literature and practical performance).
>
> #### Related Literature
>
> We appreciate the suggestions for improvement of our related literature section and have added additional references such as the mentioned SVGD variants and HMC methods.
>
> #### Practical Performance
>
> Since Sun et al. and our approach minimize the same objective function, both can be expected to perform similarly. However, the goal of our work was not to propose a method that outperforms Sun. Instead, we here derive a general gradient descent method that enables us to connect Sun's theoretical framework (and implied guarantees) with, e.g., current heuristic approaches in deep ensemble learning. Note, however, that in the course of deriving this general framework,  we also derived a method that only scales quadratically in the number of ensemble members, while the approach by Sun et al. scales cubically (because of the SSGE). This lower computational complexity results in a shorter runtime of method compared to theirs.
> We have now added a discussion to the results section of the paper, explaining the given results.
>
> #### Typos and Minor Issues
>
> Thank you for spotting the typos and ways to improve the paper’s clarity. This is now fixed in the revised version of our paper.

---

### Comment · Area_Chair_x6zD · 2022-11-15
**Please engage before the author-reviewer discussion closes**

Dear authors and reviewers,

The first phase of the discussion period is about to close on November 18.

For authors, please make sure to submit your rebuttal by the deadline. Leave some time for the reviewers to read it and respond while you are still allowed to further engage with them. Interactions between authors and reviewers are very important for the quality of the review process, so please make sure to engage.

For reviewers, please try to acknowledge and respond to the authors' rebuttal while the discussion period is still open for them to further interact with you.

Thank you for your participation in the review process!

Best,
The AC

---

### Author Response · Authors · 2022-11-18
**Summary of updates from Authors**

Dear AC and all Reviewers,

We sincerely appreciate the AC’s and all reviewers’ time and insightful comments, which helped a lot in further improving our paper. We are very glad that reviewers **NXDp** and **bTr3** even took their time to gracefully respond to us and to point out further improvements missing at the time of our first revision. After some iterations, we think that all their comments should be resolved now and would very much appreciate it if the reviewers would take our additional work and revision into account in their assessment.

We also think that we answered all comments by reviewer **qRMB**, who – similarly to the reviewers **NXDp** and **bTr3** – provided suggestions for additional references that we have included in our manuscript. The reviewer also raised further interesting technical questions which we believe are also now resolved by our response. We hope that the reviewer manages to find some time after the rebuttal phase to take a closer look at our responses as well as other active discussions. We would like to thank again the reviewer and would be very grateful if also reviewer **qRMB** could account for our revision by updating the review score.
Finally, we thank reviewer **9cu5** for the additional comments. As suggested, we removed the text colors in our paper.
### Summary
We would like to summarize our revision and point out the improvements of our manuscript:
* **[Experiments]** We added a GP baseline to our primary benchmark, showing that SFVNN can be used as a competitive, more flexible alternative to a GP (likelihood and prior process do not have to be Gaussian). Since Sun et al. and our approach minimize the same objective function, both can be expected to perform similarly. However, the goal of our work was not to propose a method that outperforms Sun et al.. Instead, we here derive a general gradient descent method that enables us to connect Sun's theoretical framework (and implied guarantees) to other fields such as gradient boosting with uncertainty and repulsive deep ensembles.

* **[Related Literature]** We extended the Stein-based literature, pointing out how our method could be further improved. We added MCMC methods to give a broader perspective on BNN approaches. We added a second recent work which also builds upon Sun et al. showing that this research field is still active and relevant. Also, we now include repulsive deep ensembles. Here, we stress that these methods and their empirical evaluations are mainly based on weight priors. Their proposed way to handle functional priors is similar to Wang et al., such that they need access to an analytical marginal density of the posterior process (we do not). Also, they do not sample measure points, an essential ingredient of our method, enabling us to maximize a true lower bound of the log marginal likelihood (in the full-batch setting). This shows that our method bears only a superficial resemblance to the update step of repulsive deep ensembles.

* **[Runtime Complexity]** We added a runtime experiment showing that the number of particle functions increases the runtime of FVBNN more substantially than SFVNN, which is also backed up by our theoretical findings. We now also highlight this aspect more prominently in the revised manuscript.

* **[Boosting Approach]** We have added a background paragraph on gradient boosting in Appendix~A.2. Lastly, we added two additional approaches (one baseline and one state-of-the-art approach) to compare with our boosting approach. Results show that SFVGB while being often better than a baseline, is often outperformed by state-of-the-art approaches (as already mentioned in the first version of our paper); however, we want to stress SVFGB still offers an interesting and novel theoretical connection between GB and Bayesian non-parametric inference.


Again, we greatly appreciate the time and efforts of reviewers and the AC and thank everyone for helping to improve our manuscript! We firmly believe that *Stein functional variational gradient descent* is a promising research direction for functional Bayesian inference and that the provided bigger picture in our paper opens up plenty of new research topics and connections.
Authors

---

### Decision · Program_Chairs · 2023-01-20

**Decision:**

Accept: poster

**Justification For Why Not Higher Score:**

This paper introduces a novel gradient descent algorithm in function space. The method is well-motivated and discussed. The experiments sufficiently support the claims of the authors. However, the research is also incremental, so I do not recommend a higher score.

**Justification For Why Not Lower Score:**

All initial concerns raised by the reviewers were addressed during the discussion.

**Metareview: Summary, Strengths And Weaknesses:**

The paper has received mixed reviews, with all 4 reviewers proposing a borderline decision (5-5-6-5). All reviewers agree that the paper's contribution for approximate Bayesian inference in function space is valuable. However, the reviewers have raised several concerns, including the clarity of the presentation, the absence of relevant recent work in the literature review, and the inconclusive results against Sun et al. To the best of my knowledge, the authors have swiftly addressed all these concerns during the discussion. For this reason, I believe the paper is now ready for publication.

**Note From Pc:**

if the above contains the word "oral" or "spotlight" please see: "oral" presentation means -> notable-top-5% and "spotlight" means -> notable-top-25%. As stated in our emails, we are disassociating presentation type from AC recommendations

**Summary Of Ac-Reviewer Meeting:**

The author-reviewer discussion and the convincing rebuttal and improvements made by the authors were sufficient to reach a conclusive decision.